# Thresholds of temperature change for mass extinctions

Haijun Song [1✉], David B. Kemp[1], Li Tian [1], Daoliang Chu [1], Huyue Song[1] & Xu Dai[1]

Climate change is a critical factor affecting biodiversity. However, the quantitative relationship between temperature change and extinction is unclear. Here, we analyze magnitudes and rates of temperature change and extinction rates of marine fossils through the past 450 million years (Myr). The results show that both the rate and magnitude of temperature change are significantly positively correlated with the extinction rate of marine animals. Major mass extinctions in the Phanerozoic can be linked to thresholds in climate change (warming or cooling) that equate to magnitudes >5.2 °C and rates >10 °C/Myr. The significant relationship between temperature change and extinction still exists when we exclude the five largest mass extinctions of the Phanerozoic. Our findings predict that a temperature increase of 5.2 °C above the pre-industrial level at present rates of increase would likely result in mass extinction comparable to that of the major Phanerozoic events, even without other, non-climatic anthropogenic impacts.

[1] State Key Laboratory of Biogeology and Environmental Geology, School of Earth Sciences, China University of Geosciences, Wuhan, China.
✉email: haijunsong@cug.edu.cn

F ive large-magnitude mass extinctions (the "Big Five") have occurred during the past 450 million years (Myr)[1], where the estimated extinction of marine animals for each event was over 75% at the species level[2]. A large body of evidence has focused on abrupt climate change (both warming and cooling) as a direct or indirect mechanism that drove many mass and minor extinctions[3–8]. Of the Big Five extinctions, for example, the end-Ordovician mass extinction (~443 Ma) was related to a short-lived cooling event accompanied by a glaciation maximum and a major drop in sea level[7,9]. The Permian-Triassic mass extinction (~252 Ma), the largest of the Phanerozoic[10], occurred within a short interval of ~60,000 years and was associated with rapid climate warming[8,11]. Although the temporal coincidence between climate change and extinction is clear, there is a paucity of quantitative analysis investigating the precise relationship between magnitudes and rates of temperature change and extinction through the Phanerozoic Eon.

Paleotemperature data for the Phanerozoic have increased in abundance enormously over the past two decades, furnishing information for each geologic stage in post-Cambrian periods[12]. These data allow investigation of the role that temperature change played in past extinctions. For this study, a paleotemperature database is constructed for 45 approximately uniform time intervals (averaging ~10 Myr) spanning the late Ordovician (~450 Ma) to the early Miocene (~15 Ma) (Supplementary Data 1). Within each time interval, the largest magnitude of temperature change and its duration and rate are quantified. Paleotemperatures of surface seawater are calculated from multiple proxies, including oxygen isotopes from carbonate and apatite fossil shells ($\delta^{18}O$), carbonate clumped isotopes ($\Delta 47$), and organic geochemical proxies such as $TEX_{86}$ (see Methods). These data are derived primarily from tropical and subtropical regions (Source Data). To quantify extinction within each of the 45 time intervals, we use two rate estimators: gap-filler (GF) extinction rate and three-timer (3 T) extinction rate, which are calculated using data from the Paleobiology Database (see Methods). We find that both the rate and magnitude of climate change are positively correlated with the extinction rate of marine animals.

## Results and discussion
### Climate change and extinctions.
Figure 1a shows the maximum magnitudes ($\Delta T$) of temperature change within each of the 45 time bins through the past 450 million years. Each time bin is defined by one or several contiguous geologic stages with an average duration of 9.71 Myr. Figure 1b shows the rates ($R$) of temperature change (Fig. 1b), where $R$ is defined as the maximum magnitude of change within a time bin (i.e., $\Delta T$) divided by the time interval over which $\Delta T$ occurs (Supplementary Fig. 1). Figure 1c shows GF extinction rates of marine animals, including the Big Five mass extinctions that occurred at the end-Ordovician, Frasnian-Famennian transition, Permian-Triassic transition, Triassic-Jurassic transition, and Cretaceous-Paleogene transition[1]. All Big Five extinction events occur within intervals associated with both high magnitudes and high rates of climate change (Fig. 1).

Correlation analysis (Spearman's $\rho$ rank correlation) reveals significant relationships, both between $\Delta T$ and GF extinction rate ($n = 45$, $\rho = 0.63$, $P < 0.001$, Fig. 2a) and between $R$ and GF extinction rate ($n = 45$, $\rho = 0.57$, $P < 0.001$, Fig. 2c). Our datasets of GF extinction rate, $\Delta T$, and $R$ are not significantly autocorrelated, and as such these correlation analysis results are not affected by the potential influence of serial correlation (see Methods). The significant correlations between climate change and biotic extinction still exist when we use 3 T extinction

rate ($n = 45$, $\rho = 0.59$, $P < 0.001$ for $\Delta T$, $n = 45$, $\rho = 0.54$, $P < 0.001$ for $R$) (Supplementary Table 1), thus providing parallel supporting evidence that the magnitude and rate of climate change are the dominant factors that can explain variations in the magnitude of marine extinctions through the Phanerozoic.

When cooling events are separated from warming events, the correlations remain. The magnitudes of warming are positively correlated with extinction ($n = 20$, $\rho = 0.76$, $P < 0.001$), and the magnitudes of cooling are negatively correlated with extinction ($n = 25$, $\rho = -0.53$, $P = 0.006$, Fig. 2b). Similarly, the rates of warming events are positively associated with extinction ($n = 20$, $\rho = 0.78$, $P < 0.001$), while the rates of cooling events have a weak and insignificant negative relationship with extinction ($n = 25$, $\rho = -0.31$, $P = 0.134$, Fig. 2d). Notably, the slopes of extinction on rate and magnitude of temperature change during warming are steeper than those during cooling (Fig. 2b, d). However, the results of a Pearson-Filon test show that there is no significant difference between the correlation coefficients of extinction rate and $\Delta T$ under warming and cooling events (Pearson-Filon's $z = 1.57$, $P = 0.12$, Supplementary Table 2). The correlation coefficients of extinction rate and $R$ under warming and cooling events are also not significantly different (Pearson-Filon's $z = 1.23$, $P = 0.22$, Supplementary Table 2).

A relationship between large-scale climate change and mass extinction is well established for the Big Five extinction events of Earth history, and it could be argued that these large-scale events bias our analysis and that the relationship between climate and smaller losses of biodiversity is equivocal. To test this, we excluded the Big Five mass extinction events from our compilation and repeated our analysis. The magnitude of temperature change still has a significant positive relationship with the GF extinction rate, but this relationship is weaker ($n = 40$, $\rho = 0.54$, $P < 0.001$). Similarly, a significant but weaker correlation was also found between $R$ and GF extinction ($n = 40$, $\rho = 0.42$, $P = 0.007$). The correlation between the magnitude of temperature change and the 3 T extinction also becomes weaker but remains significant ($n = 40$, $\rho = 0.47$, $P = 0.002$). The correlation between $R$ and 3 T extinction is weak and significant ($n = 40$, $\rho = 0.36$, $P = 0.021$) (Supplementary Table 1). We note that except for the Big Five extinctions, only one climate event (the Paleocene-Eocene thermal maximum event, ~55 Ma) has a warming rate higher than 10 °C/Myr (Fig. 3). Together, these analyses suggest that the magnitude and rate of change in temperature played a significant role in influencing the extinction rate of marine animals during the past 450 Myr, and that the rate of temperature change is a weaker influence on extinction rate if rates of change are <10 °C/Myr ($n = 39$, $\rho = 0.41$, $P = 0.009$). This finding emphasizes the existence of stress on marine organisms from a temperature change (including both the magnitude and rate), and that the impact of temperature change will be tempered if organisms have time to adapt (e.g., ref. [13]).

Most major and minor mass extinctions occurred on very short geological timespans rather than across a whole stage/series[14]. Thus, using the extinction rates calculated for bins that encompass one or several stages could, in theory, bias our analysis of the relationship between climate change and extinction. To test this, we compared the timing of major and minor mass extinctions with the timing and timespan of the largest magnitudes of climate change within the bins containing these extinction events. Within the resolution of the available data, we find that all the Big Five extinctions and five out of six minor extinctions occurred within the interval of the most significant temperature change (Fig. 4). In other words, the timing of extinction is typically contemporaneous with the most significant climate change in a given time bin. The exception to this is the end-Devonian extinction (Fig. 4), where a paucity of

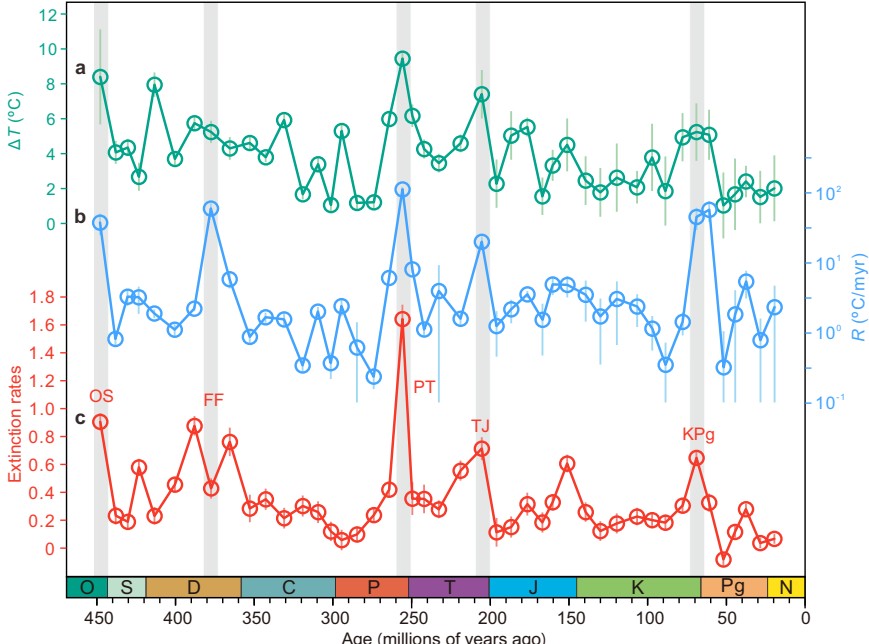

**Fig. 1 Temperature change and extinction rate over the past 450 million years. a** The largest magnitude of temperature change ($\Delta T$, absolute value) in each time bin. **b** The highest rate ($R$, absolute value) of temperature change in each time bin, defined at the million-year (Myr) scale. **c** Generic extinction rates of marine animals calculated using gap-filler methods using data from the Paleobiology Database. The Big Five extinctions occurred in the end-Ordovician (OS), Frasnian-Famennian transition (FF), Permian-Triassic transition (PT), Triassic-Jurassic transition (TJ), and Cretaceous-Paleogene transition (KPg). Vertical bars show mean ±1 x standard deviation (see Methods). O Ordovician, S Silurian, D Devonian, C Carboniferous, P Permian, T Triassic, J Jurassic, K Cretaceous, Pg Paleogene, N Neogene. Dark cyan, blue, and red dots represent $\Delta T$, $R$, and extinction rate, respectively. Source data are provided as a Source Data file.

temperature measurements in the latest Devonian precludes establishing an unequivocal link between temperature change and extinction. Nevertheless, the temporal coincidence between the latest Devonian glaciation and the end-Devonian extinction suggests a causal link between cooling and extinction[15].

**Thresholds of temperature change for mass extinctions**. Our analysis shows that for all the Big Five mass extinction events, magnitudes of temperature change ($\Delta T$) likely exceeded 5.2 °C (Fig. 3). Specifically, the Permian-Triassic mass extinction occurred during the warming of >10 °C, and at a rate (defined at the million-year timescale) of $10^2$–$10^3$ °C/Myr[8,11]. The end-Ordovician mass extinction occurred during cooling of ~8.4 °C at a rate of $10^1$–$10^2$ °C/Myr[7,16]. The Triassic-Jurassic extinction event occurred during the warming of ~7.4 °C at a rate >10 °C/Myr[17], while the Frasnian-Famennian and the Cretaceous-Paleogene mass extinctions were associated with cooling of ~5.2 °C at a rate of $10^1$–$10^2$ °C/Myr[18–20]. As noted above, the precise timescales of extinction during these events are likely to be short[11] and are hard to quantify in the geological record. As such, rate estimates such as ours cannot reveal the true pace of temperature changes that biota may have experienced across very short timescales[21]. However, we note that all these major extinctions were associated with rates of temperature change >10 °C/Myr as defined at the million-year scale, and none of the background/minor extinctions exceeds either $\Delta T$ >5.2 °C or $R$ >10 °C/Myr (Fig. 3). We suggest that a $\Delta T$ of >5.2 °C and $R$ of >10 °C/Myr likely represent the critical thresholds needed to attain a mass extinction on a par with the Big Five.

Observations in laboratory experiments and oceanographic surveys show that a significant increase/decrease in temperature can lead to mortality of marine organisms when temperatures reach upper/lower thermal limits[21]. For example, global warming

of ~1 °C has triggered mass bleaching of corals[22]. Model studies suggest that coral reef demise will be more widespread if the global mean temperature exceeds 2 °C above the pre-industrial level[23]. Climate change can also lead to diversity decline via changing species interactions[24]. The inability for species to track their preferred habitat under climate warming would also increase extinction risk[25]. Additionally, the rate of climate change plays an essential role in habitat availability. Rapid changes in temperature can lead to less time for ecosystems to adapt, and result in significant habitat loss[26,27].

Our results do not exclude the role of other factors in driving mass extinction events, such as large igneous province eruption, bolide impact, ocean acidification, and anoxia, since these factors can typically be related to climate change either as a trigger or a consequence[21,28]. For example, large igneous province emplacement can release large volumes of $CO_2$, leading to rapid global warming, e.g., the Siberian Traps and the rapid climate warming in the latest Permian[8,29]. Recent $TEX_{86}$ data show that the Cretaceous-Paleogene cooling event resulted from the complex superimposition of a long-term climate cooling in the latest Cretaceous and a short-lived cooling event at the Cretaceous-Paleogene boundary[19]. The latter event was likely caused by the "impact winter" over a time interval of months to decades following the Chicxulub impact[19]. Ocean anoxia is usually related to global warming because a rise in temperature can not only decrease dissolved oxygen levels in seawater, but also weaken the circulation of ocean currents, thus limiting the transport of oxygen from the surface to the deep ocean. Ocean anoxia is associated with warming during both the Permian-Triassic and the Triassic-Jurassic boundary events[5,8,30,31]. Anoxia as a kill mechanism can also be linked to increased metabolic demand for oxygen by animals caused by higher temperatures[28]. In addition, laboratory experiments show that

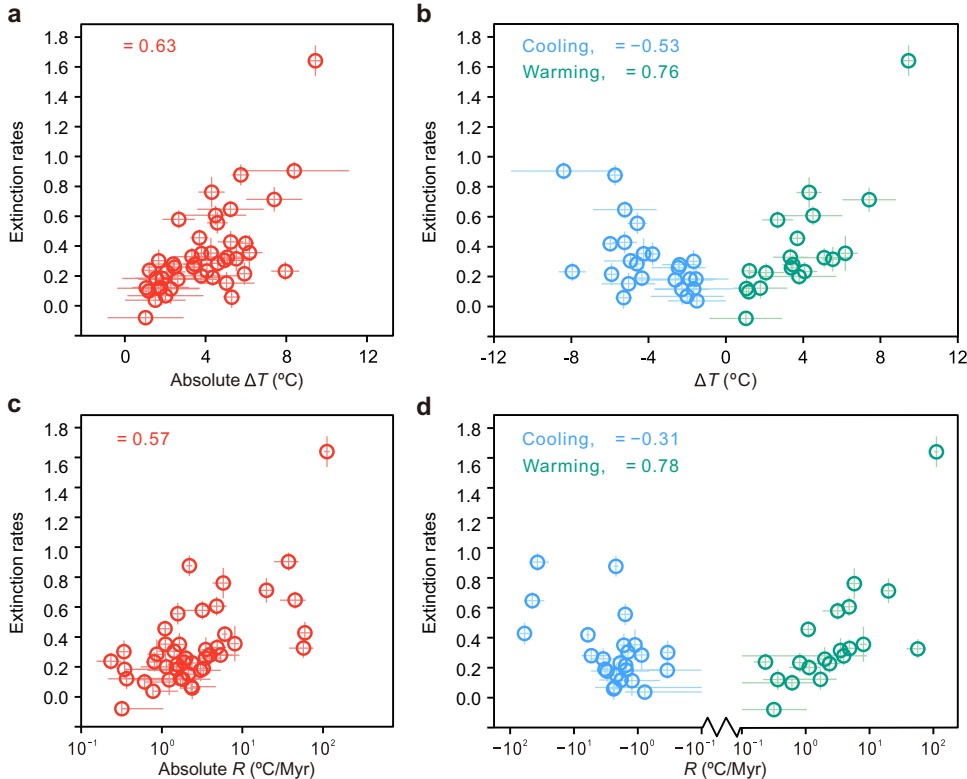

**Fig. 2 The relationship between climate change and GF extinction rate. a** Cross plot of $\Delta T$ and extinction rate ($n = 45$, $\rho = 0.63$, $P < 0.001$). After excluding the Big Five extinctions, the correlation persists ($n = 40$, $\rho = 0.54$, $P < 0.001$). **b** Cross plot of $\Delta T$ and extinction rate, with blue dots representing cooling events ($n = 25$, $\rho = -0.53$, $P = 0.006$), and green dots representing warming events ($n = 20$, $\rho = 0.76$, $P < 0.001$). **c** Cross plot of $R$ and extinction rate ($n = 45$, $\rho = 0.57$, $P < 0.001$). After excluding the Big Five extinctions, the correlation persists ($n = 40$, $\rho = 0.42$, $P = 0.007$). **d** Cross plot of $R$ and extinction rate, with blue dots representing cooling events ($n = 25$, $\rho = -0.31$, $P = 0.134$), and green dots representing warming events ($n = 20$, $\rho = 0.78$, $P < 0.001$). $n$ = the sample size used to derive statistics. Horizontal and vertical bars show mean ±1 x standard deviation (see Methods). Red, blue, and dark cyan dots represent all, cooling events and warming events, respectively. The statistical test was two-sided and no adjustments were made for multiple comparisons. Source data are provided as a Source Data file.

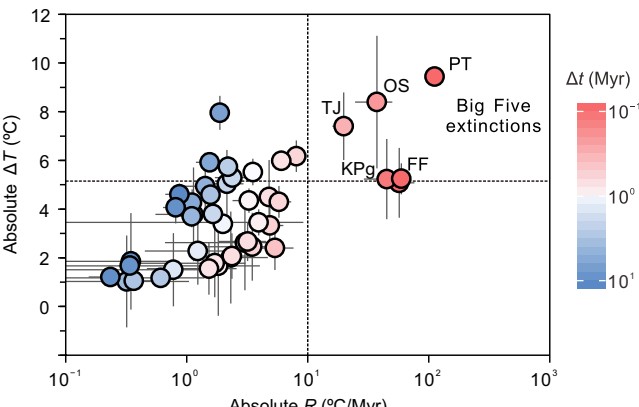

**Fig. 3 Cross plot of magnitude versus rate of temperature change.** Higher magnitude climate change tended to occur at faster rates. The Big Five extinction events fall in the area of $\Delta T > 5.2$ °C, $R > 10$ °C/Myr, and timespan ($\Delta t$) < 0.4 Myr, thus defining the broad climate thresholds that lead to mass extinction of marine animals. As made clear in the main text, the plotted magnitudes and rates pertain to those that occur within each of the 45 defined time intervals. Horizontal and vertical bars show mean ±1 x standard deviation (see Methods). The colour of dots represents the timespan of each climate event. Source data are provided as a Source Data file.

synergistic stressors of warming and hypoxia are especially lethal to marine animals[32].

Mayhew et al.[3] reported a positive relationship between Phanerozoic temperatures and extinction rates of marine invertebrates. Both intrinsic and extrinsic factors have been used to explain this correlation[3]. Marine ecosystems tended to be dominated by broadly adapted and long-lived taxa during the late Paleozoic ice age, and therefore had sluggish turnover rates[33,34]. Warm and productive shelf seas are more conducive for high marine diversity, but such settings are vulnerable to hypoxia/anoxia during a warm climate[34].

A common cause hypothesis has been used to interpret the correlation between extinction and observed environmental changes, e.g., sea level, both of which are related to the amount of exposed sedimentary rock[35,36]. However, the correlation between extinction rate and temperature change is not an artifact of variability in the amount of exposed rock because the temperature magnitude and rate are independent of variability in the stratigraphic record. Similarly, it could be argued that higher sea levels could lead to a clustering of extinctions in deep water species (e.g. ref. [32]). However, the significant correlations we observe between extinction rates and temperature change are also observed after excluding the deep-water fossil records (Supplementary Table 1).

Previous work has shown that measured magnitudes and rates of temperature change in the geological record are strongly

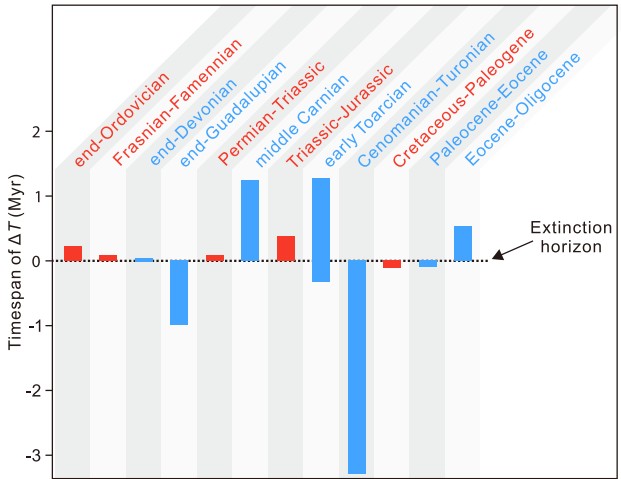

**Fig. 4 Timing of temperature change (ΔT) relative to extinction horizon for the major (red) and minor (blue) extinctions during the past 450 million years.** The dashed line is the position of the extinction horizon in relation to the time span of the largest magnitude climate changes (colored bars) within the time interval containing the extinction. The exception to this is the end-Devonian bar (see Supplementary Fig. 3). This bar represents instead the timespan of glaciation in the latest Devonian. The plot emphasizes how the timing of extinction is contemporaneous with large magnitude climate change (within the resolution of the available data). For the ages of mass extinctions and ΔT see Supplementary Table 3. Red and blue bars represent Big Five and minor mass extinctions, respectively. Source data are provided as a Source Data file.

controlled by the timespan over which rates are measured[37]. To mitigate this issue, we focused on time intervals with well-studied and reasonably high-resolution geochemistry data. Time intervals with only low-resolution paleotemperature data (i.e., < 2 measurements/Myr) are not included in our analyses, e.g., all-time bins in the Cambrian and most bins in the Ordovician. For our study, the broadly even partitioning of data into ~10 million-year time bins limits any potential bias in the extinction data, and there is no correlation between time bin duration and extinction rate ($n = 45$, $\rho = 0.14$, $P = 0.35$). This permits reasonable comparison to be made of rates of extinction between different time intervals across the Phanerozoic. In contrast, our quantification of climate change is based on the maximum magnitudes (and associated rates) of climate change within each time bin, and these magnitudes are defined over time spans ranging from 0.05 to 6.5 Myr (Source Data). Importantly, however, magnitudes of climate change do not increase with increasing timespan ($n = 45$, $\rho = -0.148$, $P = 0.331$). This is because the time intervals encompassing the Big Five mass extinctions are associated with large magnitude and rapid climate change (Supplementary Fig. 2), as noted earlier.

**Implications of climate thresholds for extinction.** There are clear resolution limitations of both the data and the geological record that, coupled with the potential biases noted above, complicate any attempt to apply the climate thresholds for extinction established in this study (defined on million-year timescales) to the rates of climate change and biodiversity loss observed at the present day[37]. In particular, as noted above, maximum rates of temperature change are underestimated in the fossil record[37], and knowledge of the peak rates of warming during Phanerozoic mass extinctions on societally relevant (decadal to millennial) timescales is not readily recoverable from geological data. Nevertheless, global mean temperatures have already risen by ~1 °C since 1850[38], and the heavy fossil fuel use

scenario trajectory of anthropogenic carbon emissions (Shared Socioeconomic Pathway, SSP5-8.5) predicts that a temperature increase matching our geologically defined magnitude threshold for mass extinction (i.e. 5.2 °C above the pre-industrial level) would be reached by ~2100[39]. The potential achievement of our defined magnitude threshold on this timescale would lead to mass extinction comparable to the major Phanerozoic events, regardless of other, non-climatic anthropogenic changes that negatively affect animal life.

## Methods

**Paleotemperature estimators.** Phanerozoic paleotemperature data are derived from oxygen isotope ($\delta^{18}O$), clumped isotope ($\Delta 47$), and organic geochemistry (TEX$_{86}$) data. Many published studies have used different conversion formulae to get from measured values to temperatures for each paleothermometry method, which makes it difficult to compare results from various studies. Because of this, we used a uniform formula to re-calculate paleotemperatures. Different estimators generally show consistent trends for a given time interval, e.g. Cretaceous planktonic foraminifera $\delta^{18}O$ and TEX$_{86}$[40], and Jurassic belemnite $\delta^{18}O$[41].

Most Paleozoic and Mesozoic paleotemperature data come from oxygen isotope paleothermometry. Cenozoic paleotemperature data derive mainly from TEX$_{86}$ and oxygen isotope of planktonic foraminifera. At present, there are major debates over the first-order trend of Phanerozoic surface temperatures calculated from oxygen isotopes[12,42–44]. For instance, it is uncertain whether $\delta^{18}O_{sw}$ evolved towards more enriched values due to exchange at mid-ocean ridges, potentially impacting interpretations of seawater temperatures in the early Paleozoic, for instance[43]. However, this is not an issue in our study because we focus only on changes in temperature rather than absolute values.

The data we used for each time bin was selected from a large paleotemperature dataset[12] and published literature by adopting the following criteria: 1) data with well-constrained ages; 2) data measured at high time resolution; 3) data from tropical/subtropical regions. For multiple sets of data in the same time bin, we use the average values. Where possible, in order to test the reliability of climate events, we use temperature data from two different proxies or two different regions. For example, $\delta^{18}O$ of planktonic foraminifera and TEX$_{86}$ were used for the late Eocene (Pg4) and Maastrichtian (K8) intervals; and conodont $\delta^{18}O$ from South China and Armenia were used for the Lopingian (P5) interval. The results show that data from two different proxies or different regions have a similar magnitude and rate of temperature change (Supplementary Fig. 4).

**Oxygen isotopes.** Oxygen isotope values from carbonate fossils (i.e., planktonic foraminifera, brachiopod, oyster, and belemnite) are converted to SSTs using the BAYFOX Bayesian model (https://github.com/jesstierney/bayfoxm)[45]. For phosphate fossil conodont, we used Monte Carlo simulations to propagate parameter uncertainties to temperature estimation by using the equation of Pucéat et al.[46]:

$$SST = 118.7(\pm 4.9) - 4.22(\pm 0.20) \times \left( \delta^{18}O_{con} - \delta^{18}O_{SW} \right) \times 1000 \qquad (1)$$

where $\delta^{18}O_{con}$ is the oxygen isotope composition of conodonts, and $\delta^{18}O_{SW}$ is the oxygen isotope composition of seawater.

Seawater oxygen isotope composition is affected by salinity and changes in continental ice volume[43,47]. As such, $\delta^{18}O$ data from localities with abnormal salinities (e.g., evaporite facies, upwelling regions) were excluded (see paleotemperature estimators above). Changes in continental ice volume were also considered in our calculation of paleotemperatures. To do this, the oxygen isotope value of seawater was set to −1‰ (VSMOW) for ice-free time intervals[48] and +1‰ (VSMOW) during glacial maximum intervals, e.g., the Pennsylvanian Glacial Maximum and the Pleistocene Last Glacial Maximum[49]. Most oxygen isotope data are from tropical and subtropical regions (Supplementary Fig. 5), therefore, no latitudinal seawater corrections were made. In addition, seawater pH has a further important influence on the $\delta^{18}O$ of foraminifera[50]. During global warming events that are related to $CO_2$ release, seawater pH can decrease, which would lead to underestimate SSTs[50,51]. Unfortunately, pH estimates for the Phanerozoic time bins are few, and their values are highly uncertain[52]. Therefore, we did not pH-correct $\delta^{18}O$ estimates. Oxygen isotope values that were likely affected by diagenetic alteration were also removed from the database. Diagenetic screening criteria included $\delta^{18}O$ values that are unrealistically negative or positive, and $\delta^{18}O$ values from carbonate fossil shells with [Mn] > 250 ppm and [Sr] < 400 ppm[17]. In addition, $\delta^{18}O$ records that showed clear offsets for a given interval among different sites were also likely influenced by diagenesis[53] or local effects (e.g. abnormal salinity as noted above) and thus not used in this study.

**TEX$_{86}$.** Previous studies have introduced several calibrations to convert TEX$_{86}$ values into SSTs, e.g., linear relationship, logarithmic relationship, and Bayesian regression models[54,55]. To get consistent and uniform data, the Bayesian spatially varying regression (BAYSPAR) approach was used to translate TEX$_{86}$ values into SSTs (https://github.com/jesstierney/BAYSPAR) calibration)[55,56].

A Branched and Isoprenoid Tetraether Index (BIT index) above 0.4 may indicate that $TEX_{86}$ values are compromised by the input of terrestrially derived branched GDGTs (glycerol dialkyl glycerol tetraethers)[57]. Therefore, data associated with BIT indices >0.4 were not used in this study. Other screening criteria of $TEX_{86}$ were also employed. Notably, data were removed from the database if Methane Index (MI) >0.5 (ref. [58]), delta-Ring Index (ΔRI) > 0.3 (ref. [59]), %GDGT-0 >67% (ref. [60]), and/or fCren':Cren' + Cren >0.25 (ref. [40]).

**Age model.** Geochronologic constraints can provide absolute age at the initiation and termination of warming/cooling events. In our database, only one warming event, at the Permian-Triassic boundary, meets this criterion. Other dates were obtained using a comprehensive approach including isotope geochronology, astrochronology, and biostratigraphy with reference to the Geologic Time Scale 2012[61]. Age data were applied in the following order of priority: isotope geochronologic age, astrochronologic age, and biostratigraphic age. If isotope geochronologic ages were available, we gave priority to these absolute dates. In the absence of absolute age data, astrochronologic ages were preferred. If neither of these numerical data was available, the climatic events were timed based on the age of biozones in the Geologic Time Scale 2012[61]. Relative ages within individual biozones were constrained based on the stratigraphic position. Age uncertainty was calculated for all events by using Monte Carlo simulation based on the assumption of uniform distribution of ages.

**The magnitude and rate of temperature change.** The maximum magnitudes (ΔT) of temperature change within each of the 45 defined time intervals were calculated as:

$$\Delta T = T_1 - T_0 \tag{2}$$

where $T_0$ and $T_1$ represent the initial and terminal temperature of warming/cooling events, respectively. Supplementary Fig. 1 illustrates how the parameters $T_0$ and $T_1$ are derived from a climatic time series $t$.

The time scale Δt represents the duration of time during which the temperature increases/decreases. Δt was calculated as:

$$\Delta t = t_1 - t_0 \tag{3}$$

where $t_0$ and $t_1$ represent the initial and terminal time of warming/cooling events, respectively.

The rate (R) of temperature change was computed as:

$$R = \Delta T / \Delta t \tag{4}$$

The ratio $R$ represents the rate of a single warming/cooling event over geological time.

ΔT, R and their uncertainties were calculated by using Monte Carlo simulation based on the distribution of data from two groups around $t_0$ and $t_1$.

**Temperature database.** The temperature database is composed of the most significant warming/cooling events in 45 time intervals from the Late Ordovician (445 Ma) to early Miocene (16 Ma) (Supplementary Data 1[62]). The time intervals are consistent with the time bins used to compute biodiversity and evolutionary rates[63,64], and are defined by one or several neighboring geologic stages with roughly uniform durations (averaging 9.71 Myr). Time bins range between 4.7 and 18.9 Myr. The 45 bins are Katian and Hirnantian (Or5), Llandovery (S1), Wenlock (S2), Ludlow and Pridoli (S3), Lochkovian and Pragian (D1), Emsian (D2), Eifelian and Givetian (D3), Frasnian (D4), Famennian (D5), Toutnaisian (C1), early Visean (C2), late Visean and Serpukhovian (C3), Bashkirian (C4), Moscovian and Kasimovian (C5), Gzhelian (C6), Asselian and Sakmarian (P1), Artinskian (P2), Kungurian and Roadian (P3), Wordian and Capitanian (P4), Wuchiapingian and Changhsingian (P5), Induan and Olenekian (T1), Anisian and Ladinian (T2), Carnian (T3), Norian (T4), Rhaetian (T5), Hettangian and Sinemurian (J1), Pliensbachian (J2), Toarcian and Aalenian (J3), Bajocian-Callovian (J4), Oxfordian (J5), Kimmeridgian and Tithonian (J6), Berriasian and Valanginian (K1), Hauterivian and Barremian (K2), Aptian (K3), Albian (K4), Cenomanian (K5), Turonian-Santonian (K6), Campanian (K7), Maastrichtian (K8), Paleocene (Pg1), Ypresian (Early Eocene, Pg2), Lutetian (early Middle Eocene, Pg3), Bartonian and Priabonian (late Middle-Late Eocene, Pg4), Oligocene (Pg5), and Aquitanian and Burdigalian (Early Miocene, Ng1).

Original proxy data, calculated temperature data, geologic age, time span, paleolatitudes, and relevant references are provided in Supplementary Methods and Source Data. Most paleotemperature data are sea surface temperatures (SST) from tropical and subtropical regions (between 40°N and 40°S, Supplementary Fig. 5). Only one collection is from a mid-latitude region with a paleolatitude of 42.71°N (Source Data). Paleolatitudes were reconstructed using PointTracker v7 rotation files published by the PALEOMAP Project[65].

For most time bins, the major climate change events (warming/cooling) were restricted to that bin. A few events occurred that cross two adjacent bins, e.g., the rapid climate warming around the Permian-Triassic and the Triassic-Jurassic boundaries[17,66,67]. Both these events had relatively short durations (61–400 kyr) and were initiated towards the end of the first (earliest) time bin. Therefore, these warmings would have primarily impacted organisms in the first bin too.

Accordingly, we didn't divide the warming event into two intervals belonging to the two adjacent bins, but take instead the warming event as belonging to the first bin.

**Extinction rate.** There are a few well-established methods to calculate extinction rate in geological history, e.g., boundary-crosser, gap-filler (GF), and three-timer (3 T) rates[63,68,69]. Both the GF and 3 T rates are calculated from occurrence data and have higher accuracy than the boundary-crosser method[64]. The boundary-crosser method is more susceptible to major biases such as the Signor-Lipps effect and sampling bias because it uses full age ranges instead of investigating sampling patterns across limited temporal windows[64]. The 3 T method can be noisy in the case of high turnover rates and poor sampling. The GF method is more precise when sampling is very poor.

Gap-filler and three-timer extinction rates of marine animals were calculated using data from the Paleobiology Database (PBDB, http://paleobiodb.org), which was downloaded on 4 January 2021. The fossil dataset includes all metazoans except for Arachnida, Insecta, Ostracoda, and Tetrapoda and consists of 850,840 fossil occurrences of 37,134 genera. These four groups that are excluded (in keeping with some previous studies[64,69]) were not used because many of them (Arachnida, Insecta, and Tetrapoda) are terrestrial and appear in marine strata. Ostracoda also has a record in terrestrial rocks. Similar to most studies using data from PBDB, we use genus-level occurrences because genus-level taxonomy is better standardized than the taxonomy of species. GF extinction rate ($r_{GF}$) is calculated by using the equation:

$$r_{GF} = \log\left(\frac{2T + pT}{3T + pT + GF}\right) \tag{5}$$

Three-timer extinction rate ($r_{3T}$) is calculated by using the equation:

$$r_{3T} = \log\left(\frac{2T_i}{3T}\right) + \log\left(\frac{3T}{3T + pT}\right) \tag{6}$$

where $GF$ represents gap-fillers, i.e., the number of genera in time bins $i - 1$ and $i + 2$ and not within bin $i + 1$; $2T_i$ represents the number of genera sampled before and within the $i$th time bin; $3T$ represents the number of genera sampled in three consecutive time bins; and $pT$ represents the number of genera sampled before and after but not within a time bin.

**Autocorrelation analysis.** We tested for serial correlation in the time series of extinction rate, ΔT, and log R, because autocorrelation could impact the calculated correlation coefficients between these data. The autocorrelation functions (ACFs) of extinction rate, ΔT, and log R are very low at all lags >1 (Supplementary Fig. 6), indicating that all-time series lack significant serial correlation.

**Reporting summary.** Further information on research design is available in the Nature Research Reporting Summary linked to this article.

## Data availability

All temperature data are available at https://github.com/haijunsong/Thresholds-of-temperature. Fossil occurrence data are available from the Paleobiology Database (http://www.paleobiodb.org). Source data are provided with this paper.

## Code availability

All scripts used to conduct analyses are available at https://github.com/haijunsong/Thresholds-of-temperature and ref. [62].

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

## Acknowledgements

We thank John Alroy for providing extinction data and Seth Finnegan for constructive comments on the early draft of the paper. We thank all contributors to the Paleobiology Database. This is Paleobiology Database contribution number 405. This study is supported by the National Natural Science Foundation of China (41821001), the State Key R&D Project of China (2016YFA0601100), and the Strategic Priority Research Program of Chinese Academy of Sciences (XDB26000000).

## Author contributions

Ha. S. conceived the study and collected the data. Ha. S. and X. D. analyzed the data. Ha. S., D. B. K., L. T., D. C. and Hu. S. contributed to the interpretation and discussion of the results. Ha.S. and D.B.K. wrote the initial draft of the manuscript and all authors contributed to subsequent revisions.

## Competing interests

The authors declare no competing interests.
