## [Peer Review File · Nature Communications]

Reviewer comments, first round –

Reviewer #1 (Remarks to the Author):

Song et al. describe an excellent analysis of a very important question, based around a new compilation of proxies of seawater temperature changes at multiple ancient events, at high temporal resolution. They highlight how those changes observed at the ‘Big Five’ mass extinctions are more severe in both rate and magnitude of climate change than those observed at ‘minor’ extinction events and other ‘background’ intervals. The topic of the paper is very important and I believe the authors have done a superb job so far in compiling, analysing and critiquing their database. I think the manuscript would be interesting for a wide audience and recommend some minor revisions which I believe could make the implications of the paper clearer, more instructive, and more robust.

(1) That the Big Five mass extinction intervals, although they are the highest rates of extinction, represent a selection that may not be ecologically meaningful, being originally defined by Raup and Sepkoski (1982) over background intervals as the significant outliers from the distribution. Thus it follows that the thresholds suggested by the authors may not be ecologically meaningful except that, if surpassed, such climate change can lead to extreme levels of extinction only observed a handful of times in the past. It is likely that humans will be direly affected not only under extreme extinction events but also at lesser levels of extinction, so I recommend that the authors aim to inform us perhaps on lesser ‘thresholds’ too. I think the threshold that the authors currently highlight should be maintained because it is a clear message, but they should at least acknowledge that a threshold set by the Big Five is not the only situation under which extinctions rates would be beyond acceptable levels.

(2) The rate and magnitude data (n = 44) being correlated are time series, and it is well known that temporally autocorrelated data increase the likelihood of spurious correlations being observed. I have no doubt that these correlations are real and significant but they may be somewhat overestimated, which may affect the balance between rate vs magnitude. Some simple steps can be made to test for this well-known issue, such as calculating Moran’s I or simply using the acf() function in R, and accounting for it, such as differencing the time series prior to a correlation test (e.g. McKinney & Oyen, 1989); I would be happy to supply R-code to do this to the authors if they need it).

*(3) Contrasting the effects of rate and magnitude of temperature change (i.e. L62-70 and 71-79) is an exceptionally interesting part of this manuscript (especially under warming), and I think more could be done here (in addition to the time series explicit analysis mentioned above). The manuscript suggests rates are more important under warming but magnitude is more important under cooling? To assess the evidence whether the differences in effect sizes (driving extinction rate) are likely to be real or not, the authors could produce confidence intervals for their correlation coefficients. Even better, the correlation coefficients could be contrasted statistically, such as using the R-package cocor or its web interface at <http://comparingcorrelations.org/> McKinney, M. L., & Oyen, C. W. (1989). Causation and nonrandomness in biological and geological time series; temperature as a proximal control of extinction and diversity. *Palaios*, 4(1), 3-15.*

L11: ‘extinction rate’ is probably more appropriate than ‘biodiversity’ here

L20: The ‘sixth mass extinction’ is already here, according to many authors, chiefly owing to the effects of other anthropogenic stressors like habitat loss and overexploitation. Thus I think the

authors could clarify here that I think they mean that climate change could cause the sixth mass extinction on its own ‘, regardless of other anthropogenic impacts’.

L41: Because this paper compares T changes with calculated with different methods, can the authors compare the methods (say, at particular time intervals where more than one method is available?) to look for any biases in the methods?

L45: change ‘mined’ to ‘calculated’ as the extinction rates are the subject here

L50: “each time bin is defined within the Paleobiology Database”. This might be best clarified; do you mean the ‘time binning’ option was selected on the download form?

L51 -56: These are a bit strange observations to be starting with and could, I think, be deleted. The absolute change during glaciations, if calculated cumulatively, would surely be dependent on number of time steps/samples i.e. the number of ups and downs being accumulated. Is this corrected for? Perhaps the non-mass extinction times are not studied to as high temporal resolution, while the opposite is true for mass extinctions, the effects of which could be mentioned at the end of this paragraph

L66-70: These sentences are a bit redundant. Also L86-87. It would be better to just choose one ‘best’ extinction rate estimator (e.g. L91, and I would also support GF rates being superior) and show correlations with others e.g. 3T only as validation in the supplementary analyses.

L76-77: As mentioned above, “slopes of extinction on rate and magnitude of temperature change during warming are steeper than”. This can be quantified by using a test that compares correlation coefficients, such as in cocor, which can tell you if you have enough difference to support this statement and others

L90-92: Perhaps change the wording here to be more cautious. So GF is not biased in this way? I am quite sure it still is, albeit less so, because of its design. Ultimately, we will never know which is more accurate.

L94: Not just ‘magnitude’, but ‘magnitude and rate of change’, isn’t it?

L96: ‘the rate of temperature change is a weaker influence on extinction rate if rates of change are <10 °C/Myr’. Unless talking about warming when it may be more important than magnitude, based on the results above? Comparing the correlations properly would be useful here

L97-99: “time to adapt”. Although adaptation is only one possible response e.g. range shift or acclimation are others

‘Temperature stress thresholds’ is vague and begs more detail. Do you also mean temperature related stressors such as deoxygenation, and hypercapnia and acidification that rise with CO₂ levels? Do you mean magnitude or rate here? Or both? Surely a more rapid rate gives less time to adapt, by definition?

L101: The following reference would be better here than reference 15: Foote, M. (2005). Pulsed origination and extinction in the marine realm. *Paleobiology*, 31(1), 6-20.

L123-5. Reference 28 (Kemp et al. 2015) should be added to this sentence.

L129: There are other ‘major marine mass extinction’s. The original definition of mass extinction needs remembering here, that these are only the ones that clearly exceed the standard distribution of extinction rates. This definition may not have ecological definition, simply being the upper values of a continuum. Perhaps the authors could write that these are the values/thresholds needed to attain a mass extinction on par with the Big Five.

L147-8: It may also be worthwhile here (also in line 152) noting that laboratory experiments show the combination of warming and hypoxia to be especially lethal, synergistic stressors under climate change. This combination is the most likely combination of climate change stressors to drive extinctions at warming events and specifically at several of the events focused on here. Reddin, C. J., Nätscher, P. S., Kocsis, Á. T., Pörtner, H. O., & Kiessling, W. (2020). Marine clade sensitivities to climate change conform across timescales. *Nature Climate Change*, 10(3), 249-253.

L156-7: Sounds strange going directly from talking about rates of temperature change to rates of extinction. Perhaps finish talking about the former before talking about the latter.

L160-7: The discussion about observed magnitudes of climate change and timespan are welcome here. Could the authors also devote a line or two to discussing the relationship between observed rates of climate change and timespan? Ref. 28 suggests that rates are systematically underestimated: that is, if you dissect a time interval into higher resolution, you will observe greater rates of change. Is this also observed in the author's data? Lines 202-206 for the Methods could be added here to this discussion. I think the steps taken by this study (Lines 202-206) are useful and mitigation of this issue is the only possible approach, since we know that even the highest resolution fossil studies are long time frames from a human perspective. Being open and honest about this issue would be best e.g. do higher resolutions tend to see higher temperature change in these data?

L172: Perhaps also add ref 28 here

L175: The common reference to RCP 8.5 as 'business as usual' has some controversy to it, e.g. see Hausfather & Peters, 2020 vs. Schwalm et al. 2020. To avoid this controversy, RCP 8.5 should probably be referred to as a 'worst case scenario' or 'heavy fossil fuel use scenario'.

Hausfather, Z., Peters, G. P. (2020). Emissions—The “business as usual” story is misleading. *Nature* 577, 618–620.

Schwalm, C. R., Glendon, S., & Duffy, P. B. (2020). RCP8.5 tracks cumulative CO2 emissions. *Proceedings of the National Academy of Sciences*, 117(33), 19656-19657.

L176: “would be reached under RCP 8.5”. See above references; this trajectory may be unlikely these days, although it is still a useful exercise to use it. But the conditional ‘would’ rather than ‘will’ should be used.

L177-9: Again, that maximum rates are underestimated in fossil record needs acknowledging here, and we will never be able to know the peak rate of change at the Permian-Triassic, for times scales as we record for today, for example.

L192: “we used the same temperature estimator for a single time interval” not clear what this means. Used a single temp estimator for a single interval?

L209: ‘moving average/smooth curves’. What is the method of smoothing? What is the window size (generally 5 data points, I believe, but 11-point in Supplementary Fig. 12?) and symmetry of the moving average?

L211: This line is not clear enough for reproducibility. What were the criteria for selection?

L213: Planktonic

L242: Please include or refer to the methodological details to permit replication of this ‘comprehensive approach’.

L255: the range of duration values would also be useful here. To be clear, are the ‘time bins’ referred to here the ones defined by the authors or the ones defined by the Paleobiology Database?

L256: “that exactly match”. ‘that’ is omitted.

L256: As in the abstract, this line states “biodiversity and evolutionary rates” but it seems that this paper only uses extinction rates. If so, just say “extinction rates”.

Fig. 2. There are points in the figures that appear green to me but they are described as orange. Please check.

Fig 4 caption. The position of the word ‘relative’ adds confusion: ‘Timing of temperature change relative to the extinction horizon’ would be more clear. Then delete ‘relative’ in next sentence.

Supplementary Materials

In the paragraph titled ‘P4, Wordian and Capitanian (268.8-259.8 Ma).’, perhaps delete the ‘had’: ‘which would have had contributed’

In 'P5, Wuchiapingian and Changhsingian (259.8-251.9 Ma)', 'Armenia' rather than 'Armenian'
Some additional thoughts that I would be interested in knowing, but I would not require the authors to address if they did not want to, are: Do the slopes in Fig2 A and C (or with warming and cooling times separated) vary systematically with geological age (does the system change over time?) or with paleolatitude of the studied region (does the sampling location systematically bias the results)? A multiple GLS regression would best to ask this question, and the partial residuals could be plotted so show any significant relationship

I am happy for the authors to contact me on carl.reddin@mfn.berlin if they have any questions regarding my review.

*Best wishes,
Carl J Reddin*

Reviewer #2 (Remarks to the Author):

General comments

This is an interesting manuscript that deals with a very actual topic: the role of temperature changes in past extinction events. The authors correlate two datasets, temperature change and extinction rates, spanning almost the entire Phanerozoic, and find a high, statistically significant, correlation between peaks of extinctions and high rates of temperature change. This results brings the authors to the conclusion that the main past extinction of the past are linked to a certain threshold of temperature change.

The idea of the study and the conclusions are interesting and the manuscript is, overall, well written. However, I have a few concerns on the way the two datasets are built, and the way analyses are run. These concerns are summarized here, and motivated in more detail below.

First, paleo-temperature dataset. I really appreciate that the authors built their own dataset by using data derived from the literature. This part of the methods is very well explained (e.g., equations used to get temperature values, how isotopic composition of seawater was estimated, and so on). However, after a look at the supplementary material, I realized that for every time bin, data derive only from one, maximum 2, published papers, from only one region-area, if not locality. This means that the final curve is a biased, regional representation of temperature changes – not a global curve. This should be clearly stated in the methods, or either corrected with the addition of more sites in the dataset.

Second, fossil dataset. It is not clear at all how the dataset was built. This part of the methods section is very short. Did the authors built their own dataset downloading data from the PBDB? Have they used published data? Have they used a dataset at the species, genus or family level? Does the dataset include data only from marine invertebrates? The authors only state that marine diversity is explored, of what taxa? This is never mentioned in the text. Furthermore, the authors do not explore changes in marine biodiversity, they explore changes in extinction rates, which is very different. Biodiversity (e.g., alpha diversity or richness) is the result of the combination of origination and extinction rates at a certain point in time. This concept should be re-considered throughout the manuscript, as certain sentences are misleading (e.g, line 11 in the abstract.

Third, the two datasets have been correlated using Spearman's rank correlation, the non-parametric version of the Pearson product-moment correlation. However, when working with the analyses of time series, even if with such simple tests, it is necessary to test for autocorrelation between the two datasets. If so, simple transformation are needed (e.g., differentiating the data).

Besides these three main concerns, which are methodological, I also think that the authors, when discussing their results, should have critically discussed results obtained by similar studies (e.g., Mayhew al. 2012, cited in the introduction) and overall, literature dealing with the interpretation of macroevolutionary patterns and extinctions. Mayhew al. 2012 for instance, clearly state that, even if they found that high extinction rates are correlated with high temperatures, to infer causality is more equivocal. The author mention in the discussion (lines 138-152) that other factors rather than temperature could have caused mass extinction events (volcanic eruptions, bolide impacts and so on), but forget another important point, which come before all of these interpretations, which is related to the structure of the fossil – and geochemical - record. Please see Peters and Foote 2020 (Nature) on the common-cause hypothesis.

Said all that, I believe that only once the authors have revised all these points, the manuscript can be considered for publication.

Specific points

11. This is misleading. The authors are not studying changes in marine fossil biodiversity, they are studying how extinction rates change through time. This is not marine biodiversity (e.g., number of species through time). Biodiversity depends on the relationship between origination and extinction rates in the same time-interval. So, this sentence is incorrect.

14. Are the authors dealing with marine animals in general (from invertebrates to vertebrates) or only with marine vertebrates? This should be stated at least one time in the text.

43-45. As stated below, in the section concerning the methods, it is not clear what kind of data have been analysed (species, genus, family level?).

78-79. As stated above, high extinction rates do not necessarily imply a biodiversity loss, if origination rates are high. This sentence should be reworded.

90-92. The differences between the two methods used to measure extinction rates should be better explained.

103-107. How did you define which the minor extinction events are? Which is the reference publication? Some authors could argue that some of the extinctions that you point out here, e.g, the Cenomanian-Turonian, are not true extinctions but a stratigraphic artefact (e.g. Gale et al. 2000 Journal of the Geological Society of London; Smith et al. 2001 Paleobiology). The reason why I am saying this is that, for instance, the Cenomanian-Turonian extinction, like others (e.g., early Toarcian, late Ordovician), are related to major sea level rises, which would cluster last occurrences of deep water species, making the rate of extinction higher. Such transgressions are presumably related to warming events, and it becomes than difficult to understand what is the main cause of the extinction (please see Holland 2010 Palaeontology and Peters and Foote 2002 Nature: the common case hypothesis).

130-137. The examples reported here refer to laboratory experiments which cannot be compared with at which temperature changed through extinction events. I would rather support the results with other examples related, for instance, to the impossibility of species to track their preferred habitat (e.g., Sunday et al. 2012 Nature Climate Change).

Figure 2. In a and c, are you using the absolute values of ΔT and R ? If so, explain it in the caption.

Also double check colors (they do not match between figure and caption).

Methods

Paleotemperature estimators: It has been a very good idea to select the studied sample by paleo-latitude, including only data coming from tropical and subtropical regions. What is lacking, and would be added, is the range of environments from which the data come from: shallow subtidal, deep subtidal (below or above storm wave base), shallow carbonate reef? I believe that this information should be added.

Temperature Database & Supplementary Dataset: I had a quick look at the dataset, and have a couple of concerns. First, the dataset is not complete, some time intervals are not listed. Missing time intervals are: T1, J4, J5, J6, K1, K2, k3, k8, Pg2, Pg3, Pg4. Second, from most of the time intervals, data come only from 1 publication and/or 1 locality. If we focus on the Jurassic, for instance, in J1, data come only from ref. Korte and Heselbo 2011, which are all sections from the same locality in UK; for J2-J3 data come only from refs. Suan et al. 2008, 2010, which are all sections from Portugal. However, there are other published papers with published stable isotope oxygen data from shells from the early Jurassic from other localities in the Palaeotethys, which have not been included, why? (e.g., Saalen et al. 1996, Palaios, Mc Arthur et al. 2000, Dera et al. 2009 EPSL, Danise et al. 2019 Palaeo 3, Ulmann et al 2020 Scientific Reports). I supposed that this would be the case also for other time-intervals.

Extinction rates: This is the most deficient part of the methods. The authors use two methods to estimate of extinction rates, the gap-filler (GF) and three-timer (3T), used for the first time by Alroy 2014. I do not understand if they made their own estimates downloading data from the PBDB database, or if they used the estimates of Alroy 2014. In the first case, it is not stated if they are using data at the species, genera, or family level, and what higher taxonomic levels they have included in the datasets (only marine invertebrates? Marine vertebrates too?). Moreover, have they included marine taxa from all the marine environments or excluded, for instance, deep water taxa that were potentially less affected by SST changes? In the second instance, if they used the data from Alroy, unfortunately they are using a dataset that needs to be updated, as the PBDB is continuously updated with new data and mistakes are continuously corrected. Moreover, the authors should state why they decided to use these 2 estimates of extinction rates (for instance, Alroy in his 2014 paper use 3 different estimates).

It would be very useful, for each bin (or at least for each Period), to have a map showing from which locations the paleobiological data come from, and from which location the paleotemperature data come from. This would help to understanding the structure of the dataset, and would highlight possible biases.

Reviewer #3 (Remarks to the Author):

Although the premise of the MS is very interesting, timely and of broad interest, at present I find some of the methods to be inappropriate and lacking adequate description in both the main body text and in the methods.

Three key issues:

1. The extinction metrics are not well described in the main body text, there is more detailed info in the methods, but a simple explanation of what each method is trying to do would really help the reader with the flow of the MS.

2. The use of "uniform calibrations" to proxy data. This is one of my biggest concerns. The reason that many different T calibration methods are applied across this vast literature is because it is necessary. By using only one oxygen isotope T equation, for example, much of the complexity of the sources and biases on these data are overlooked. Also, the choice of d18O eqn seems odd. Why this paleo temp equation? The calibration range of this is not sufficient to accommodate very warm worlds, such as the Cretaceous, and the calibration of d18O is particularly non-linear towards higher temps e.g. 30 degC +. Surely something like Kim and O'Neill with inorganic calcite over a very wide T range would have been a better choice for this? I am not an expert in all of the T proxies, so I cannot comment on whether this "uniform calibration" approach is suitable for all of the proxies listed, but do not think it is suitable for d18O even when you are just trying to capture magnitudes of change rather than explicit values.

3. My final main concern is with the age control. It is central to the approach, which is attempting to tie T change with extinction events and yet the methodology is very brief, to the extent that I would not be able to reproduce what they did based on the information given. "In our database, only one warming event, at the Permian-Triassic boundary, meets this criterion. Other dates were obtained using a comprehensive approach including isotope geochronology, astrochronology, and biostratigraphy with reference to the Geologic Time Scale 2012" How were these used/integrated? Was there a rank order of what was the most reliable? What if the different archives didn't give the same dates, how did you reconcile this? Were there any conflicting archives? Dating is non-trivial, and this seems to only get a cursory mention?

The latter two points must be better explained and justified for this current approach to stand up to further scrutiny.

Other minor comments on the MS.

Note: original reviewer comments in black; responses of authors in blue

REVIEWER COMMENTS

Reviewer #1 (Remarks to the Author):

Song et al. describe an excellent analysis of a very important question, based around a new compilation of proxies of seawater temperature changes at multiple ancient events, at high temporal resolution. They highlight how those changes observed at the 'Big Five' mass extinctions are more severe in both rate and magnitude of climate change than those observed at 'minor' extinction events and other 'background' intervals. The topic of the paper is very important and I believe the authors have done a superb job so far in compiling, analysing and critiquing their database. I think the manuscript would be interesting for a wide audience and recommend some minor revisions which I believe could make the implications of the paper clearer, more instructive, and more robust.

Response: We appreciate the reviewer's very positive comments.

(1) That the Big Five mass extinction intervals, although they are the highest rates of extinction, represent a selection that may not be ecologically meaningful, being originally defined by Raup and Sepkoski (1982) over background intervals as the significant outliers from the distribution. Thus it follows that the thresholds suggested by the authors may not be ecologically meaningful except that, if surpassed, such climate change can lead to extreme levels of extinction only observed a handful of times in the past. It is likely that humans will be direly affected not only under extreme extinction events but also at lesser levels of extinction, so I recommend that the authors aim to inform us perhaps on lesser 'thresholds' too. I think the threshold that the authors currently highlight should be maintained because it is a clear message, but they should at least acknowledge that a threshold set by the Big Five is not the only situation under which extinctions rates would be beyond acceptable levels.

Response: Thanks. We agree with your viewpoint. We agree that the thresholds of temperature change we define are not the only situation under which extinctions would happen. Other extreme situations can also cause severe extinctions. For clarity, we have revised our final paragraph to note this key point.

(2) The rate and magnitude data ($n = 44$) being correlated are time series, and it is well known that temporally autocorrelated data increase the likelihood of spurious correlations being observed. I have no doubt that these correlations are real and significant but they may be somewhat overestimated, which may affect the balance between rate vs magnitude. Some simple steps can be made to test for this well-known issue, such as calculating Moran's I or simply using the `acf()` function in R, and accounting for it, such as differencing the time series prior to a correlation test (e.g. McKinney & Oyen, 1989); I would be happy to supply R-code to do this to the authors if they need it).

Response: Added. We have performed stationarity tests for the time series of extinction rates, absolute ΔT , and absolute R . There is no trend for absolute ΔT (Mann-Kendall test, $P = 0.07$) and absolute R (Mann-Kendall test, $P = 0.82$). Extinction rates have a decreasing trend (Mann-Kendall test, $p = 0.01$). To remove the effect of autocorrelation, we used First Difference Method. The results show that the correlations between differenced extinction rates and differenced ΔT ($r = 0.49$, $P < 0.001$) and Diff R ($r = 0.51$, $P < 0.001$) are still significant (Supplementary Fig. 24). The results of autocorrelation test have been added in the Methods.

(3) Contrasting the effects of rate and magnitude of temperature change (i.e. L62-70 and 71-79) is an exceptionally interesting part of this manuscript (especially under warming), and I think more could be done here (in addition to the time series explicit analysis mentioned above). The manuscript suggests rates are more important under warming but magnitude is more important under cooling? To assess the evidence whether the differences in effect sizes (driving extinction rate) are likely to be real or not, the authors could produce confidence intervals for their correlation coefficients. Even better, the correlation coefficients could be contrasted statistically, such as using the R-package cocor or its web interface at <http://comparingcorrelations.org/> McKinney, M. L., & Oyen, C. W. (1989). Causation and nonrandomness in biological and geological time series; temperature as a proximal control of extinction and diversity. *Palaios*, 4(1), 3-15.

Response: Added. We performed Pearson-Filon tests for the correlation coefficients using the R-package cocor, as suggested. The results show that there is no significant difference between correlation coefficients of extinction rate and ΔT under warming and cooling events (Pearson-Filon's $z = 1.26$, $P = 0.21$). The correlation coefficients of extinction rate and R under warming and cooling events are also not significantly different (Pearson-Filon's $z = 1.34$, $P = 0.18$). We have revised relevant discussions in the manuscript. See Lines 73-77.

L11: 'extinction rate' is probably more appropriate than 'biodiversity' here

Response: Replaced. "biodiversity" was replaced by "extinction rate".

L20: The 'sixth mass extinction' is already here, according to many authors, chiefly owing to the effects of other anthropogenic stressors like habitat loss and overexploitation. Thus I think the authors could clarify here that I think they mean that climate change could cause the sixth mass extinction on its own ' , regardless of other anthropogenic impacts'.

Response: We appreciate this suggestion. We have added the words "regardless of other non-climatic anthropogenic impacts" in the text (last paragraph).

L41: Because this paper compares T changes with calculated with different methods, can the authors compare the methods (say, at particular time intervals where more than one method is available?) to look for any biases in the methods?

Response: Added. In order to test the reliability of climate events, where possible we added the comparison of temperature data from two different proxies or two different regions. For example,

$\delta^{18}\text{O}$ of plankton foraminifera and TEX_{86} were used for the late Eocene, Turonian-Santonian, and Maastrichtian intervals; and conodont $\delta^{18}\text{O}$ from South China and Armenia were used for the Lopingian interval. The results show that data from two different proxies or different regions show a similar magnitude and rate of temperature change (see Supplementary Fig. 22).

L45: change 'mined' to 'calculated' as the extinction rates are the subject here

Response: Changed. 'mined' has been replaced by 'calculated'.

L50: "each time bin is defined within the Paleobiology Database". This might be best clarified; do you mean the 'time binning' option was selected on the download form?

Response: We defined the time bin. The definition of time bins is given in the Methods section: "Time intervals are defined by one or several neighboring geologic stages with roughly uniform durations (averaging 9.7 Myr) exactly match, which are similar with the time bins used to compute biodiversity and evolutionary rates^{57,58}. All time bins range between 4.7 and 18.9 Myr.". For clarity, we have added more information in the main text

L51 -56: These are a bit strange observations to be starting with and could, I think, be deleted. The absolute change during glaciations, if calculated cumulatively, would surely be dependent on number of time steps/samples i.e. the number of ups and downs being accumulated. Is this corrected for? Perhaps the non-mass extinction times are not studied to as high temporal resolution, while the opposite is true for mass extinctions, the effects of which could be mentioned at the end of this paragraph

Response: Deleted. The sentences related to glaciations has been deleted.

L66-70: These sentences are a bit redundant. Also L86-87. It would be better to just choose one 'best' extinction rate estimator (e.g. L91, and I would also support GF rates being superior) and show correlations with others e.g. 3T only as validation in the supplementary analyses.

Response: Thanks. Some redundant words have been deleted, i.e., "The extinction rates from both estimators are significantly correlated with ΔT as well as R ". We leave the 3T for comparison between different extinction estimators since readers will always want to see whether using different estimators can get the same results.

L76-77: As mentioned above, "slopes of extinction on rate and magnitude of temperature change during warming are steeper than". This can be quantified by using a test that compares correlation coefficients, such as in cocor, which can tell you if you have enough difference to support this statement and others

Response: Added. We performed Pearson-Filon tests for the correlation coefficients using the R-package cocor. The results show that there is no significant difference between correlation coefficients of extinction rate and ΔT under warming and cooling events (Pearson-Filon's $z =$

1.26, $p = 0.21$). The correlation coefficients of extinction rate and R under warming and cooling events are also not significantly different (Pearson-Filon's $z = 1.34$, $P = 0.18$). We have revised the relevant discussion in the manuscript. See Lines 73-77.

L90-92: Perhaps change the wording here to be more cautious. So GF is not biased in this way? I am quite sure it still is, albeit less so, because of its design. Ultimately, we will never know which is more accurate.

Response: Revised. We added more words to introduce the methods “There are a few methods to calculate extinction rate in geological history, e.g., boundary-crosser, gap-filler (GF) and three-timer (3T) rates ^{59,64,65}. Both the GF and 3T rates are calculated from occurrence data and have a higher accuracy than the boundary-crosser method ⁵⁹. The boundary-crosser method is more susceptible to major biases such as the Signor-Lipps effect and sampling bias because it uses full age-ranges instead of investigating sampling patterns across limited temporal windows ⁵⁹. The 3T method can be noisy in the case of high turnover rates and poor sampling. The GF method is more precise when sampling is very poor.” (see Lines 333-339).

L94: Not just ‘magnitude’, but ‘magnitude and rate of change’, isn’t it?

Response: Added. “and rate” has been added here.

L96: ‘the rate of temperature change is a weaker influence on extinction rate if rates of change are <10 °C/Myr’. Unless talking about warming when it may be more important than magnitude, based on the results above? Comparing the correlations properly would be useful here

Response: Thanks. We added details of correlations here.

L97-99: “time to adapt”. Although adaptation is only one possible response e.g. range shift or acclimation are others ‘Temperature stress thresholds’ is vague and begs more detail. Do you also mean temperature related stressors such as deoxygenation, and hypercapnia and acidification that rise with CO₂ levels? Do you mean magnitude or rate here? Or both? Surely a more rapid rate gives less time to adapt, by definition?

Response: Thanks for your suggestion. Here we mean the magnitude and rate. This has been clarified in the revised version of the manuscript: “This finding emphasizes the existence of stress on marine organisms from temperature change (including both the magnitude and rate), and that the impact of temperature change will be tempered if organisms have time to adapt (e.g., ref. 13)”, see Lines 93-96.

L101: The following reference would be better here than reference 15: Foote, M. (2005). Pulsed origination and extinction in the marine realm. *Paleobiology*, 31(1), 6-20.

Response: Replaced. We use Foote 2005 in the revised version.

L123-5. Reference 28 (Kemp et al. 2015) should be added to this sentence.

Response: Added. The Kemp et al. 2015 reference has been added here.

L129: There are other ‘major marine mass extinction’s. The original definition of mass extinction needs remembering here, that these are only the ones that clearly exceed the standard distribution of extinction rates. This definition may not have ecological definition, simply being the upper values of a continuum. Perhaps the authors could write that these are the values/thresholds needed to attain a mass extinction on par with the Big Five.

Response: Revised. Here “major marine mass extinction” has been replaced by “mass extinction on a par with the Big Five”.

L147-8: It may also be worthwhile here (also in line 152) noting that laboratory experiments show the combination of warming and hypoxia to be especially lethal, synergistic stressors under climate change. This combination is the most likely combination of climate change stressors to drive extinctions at warming events and specifically at several of the events focused on here. Reddin, C. J., Nätscher, P. S., Kocsis, Á. T., Pörtner, H. O., & Kiessling, W. (2020). Marine clade sensitivities to climate change conform across timescales. *Nature Climate Change*, 10(3), 249-253.

Response: Added. The synergistic stressors of warming and hypoxia as well as the reference (Reddin et al., 2020) have been added in the manuscript (see Lines 151-152).

L156-7: Sounds strange going directly from talking about rates of temperature change to rates of extinction. Perhaps finish talking about the former before talking about the latter.

Response: Revised. We added more discussion about the rate of temperature change before talking about the extinction, i.e., “To mitigate this issue, we focused on time intervals with well-studied and reasonably high-resolution geochemistry data. Time intervals with only low-resolution paleotemperature data (i.e., < 2 measurements/Myr) are not included in our analyses, e.g., all time bins in the Cambrian and most bins in the Ordovician.” (see Lines 169-173).

L160-7: The discussion about observed magnitudes of climate change and timespan are welcome here. Could the authors also devote a line or two to discussing the relationship between observed rates of climate change and timespan? Ref. 28 suggests that rates are systematically underestimated: that is, if you dissect a time interval into higher resolution, you will observe greater rates of change. Is this also observed in the author’s data? Lines 202-206 for the Methods could be added here to this discussion. I think the steps taken by this study (Lines 202-206) are useful and mitigation of this issue is the only possible approach, since we know that even the highest resolution fossil studies are long time frames from a human perspective. Being open and honest about this issue would be best e.g. do higher resolutions tend to see higher temperature change in these data?

Response: Thanks for your suggestion. Yes, our results agree with the previous observations (Kemp et al., 2015) that higher resolutions tend to see higher rate of temperature change. Words in Lines 202-206 (Methods) have been moved here to address this issue. As your suggestion, we are open and honest about this issue and added more words “In particular, as noted above, maximum rates of temperature change are underestimated in the fossil record³³, and knowledge of the peak rates of warming during Phanerozoic mass extinctions on societally relevant (decadal to millennial) timescales is not readily recoverable from geological data.” (see Lines 189-192).

L172: Perhaps also add ref 28 here

Response: Added. The ref of Kemp et al. 2015 has been added here.

L175: The common reference to RCP 8.5 as ‘business as usual’ has some controversy to it, e.g. see Hausfather & Peters, 2020 vs. Schwalm et al. 2020. To avoid this controversy, RCP 8.5 should probably be referred to as a ‘worst case scenario’ or ‘heavy fossil fuel use scenario’.

Hausfather, Z., Peters, G. P. (2020). Emissions—The “business as usual” story is misleading. *Nature* 577, 618–620.

Schwalm, C. R., Glendon, S., & Duffy, P. B. (2020). RCP8.5 tracks cumulative CO2 emissions. *Proceedings of the National Academy of Sciences*, 117(33), 19656-19657.

Response: Thanks. “business as usual” was replaced by “heavy fossil fuel use scenario”.

L176: “would be reached under RCP 8.5”. See above references; this trajectory may be unlikely these days, although it is still a useful exercise to use it. But the conditional ‘would’ rather than ‘will’ should be used.

Response: Revised. We changed “will” to “would”.

L177-9: Again, that maximum rates are underestimated in fossil record needs acknowledging here, and we will never be able to know the peak rate of change at the Permian-Triassic, for times scales as we record for today, for example.

Response: Added. The acknowledgment of the underestimation of the maximum rate of temperature change in fossil records was added in this paragraph (see Lines 189-192).

L192: “we used the same temperature estimator for a single time interval” not clear what this means. Used a single temp estimator for a single interval?

Response: Yes, we used a single temperature estimator for a single interval. For clarity, this issue has been revised in the manuscript.

L209: ‘moving average/smooth curves’. What is the method of smoothing? What is the window size (generally 5 data points, I believe, but 11-point in Supplementary Fig. 12?) and symmetry of the moving average?

Response: Thanks. We have added details of the methods used for the moving average and smoothing calculations. We used five-point moving average curves for most time bins (see Supplementary Figs. 1-11, 13-17). For the Albian bin, we used eleven-point moving average (Supplementary Fig. 12) because this bin has high-resolution but scattered oxygen isotope data. For the Middle and Late Jurassic bins, we used the well-known paleotemperature curve established using belemnite oxygen isotopes (Dera et al. 2011). This smoothed curve is generated using Kernel regression with a bandwidth of 0.5 Myr (Dera et al. 2011). For the Eocene bins, we used smoothed paleotemperature curve established based on multiple proxies including TEX₈₆, $\delta^{18}\text{O}$ and Mg/Ca of planktonic foraminifer (Cramwinckel et al., 2018). The Eocene SST curve is generated using a local regression model with a bandwidth of <0.5 Myr (Cramwinckel et al., 2018).

L211: This line is not clear enough for reproducibility. What were the criteria for selection?

Response: Revised. We added more text in the Methods to introduce the criteria for selection, i.e. “For multiple sets of data in the same time bin, we use the one with well-constrained age (see Age model) and higher resolution data. In order to test the reliability of climate events, where possible we use temperature data from two different proxies or two different regions. For example, $\delta^{18}\text{O}$ of plankton foraminifera and TEX₈₆ were used for the late Eocene, Turonian-Santonian, and Maastrichtian intervals; and conodont $\delta^{18}\text{O}$ from South China and Armenia were used for the Lopingian interval. The results show that data from two different proxies or different regions have a similar magnitude and rate of temperature change (Supplementary Fig. 22).” (see Lines 242-249). “Age data were applied in the following order of priority: isotope geochronologic age, astrochronologic age, and biostratigraphic age. If isotope geochronologic ages were available, we gave priority to these absolute dates. In the absence of absolute age data, astrochronologic ages were preferred. If neither of these numerical data was available, the climatic events were timed based on the age of biozones in the Geologic Time Scale 2012. The relative age within the same biozone was constrained based on the stratigraphic position.” (see Lines 284-289)

L213: Planktonic

Response: Revised.

L242: Please include or refer to the methodological details to permit replication of this ‘comprehensive approach’.

Response: Added. More methodological details (e.g., how the parameters R , ΔT , Δt are derived from climatic time series’) were added in this section (see Lines 294-302).

L255: the range of duration values would also be useful here. To be clear, are the ‘time bins’ referred to here the ones defined by the authors or the ones defined by the Paleobiology Database?

Response: Revised. The range of duration values was added here. The definition of time bins followed previous studies on the evolutionary rates by Alroy et al. (2008; 2014). For clarity, this information has been added.

L256: “that exactly match”. ‘that’ is omitted.

Response: Added.

L256: As in the abstract, this line states “biodiversity and evolutionary rates” but it seems that this paper only uses extinction rates. If so, just say “extinction rates”.

Response: Revised. We have changed the word “biodiversity” to “extinction rate” in the abstract. But in this sentence “Time intervals are defined by one or several neighboring geologic stages with roughly uniform durations (averaging 9.71 Myr) that exactly match the time bins used to compute biodiversity and evolutionary rates defined by Alroy et al.”, the words “biodiversity and evolutionary rate” are kept here because Alroy et al. (2008; 2014) used these time bins to compute diversity and extinction/origination rates.

Fig. 2. There are points in the figures that appear green to me but they are described as orange. Please check.

Response: Thanks, revised.

Fig 4 caption. The position of the word ‘relative’ adds confusion: ‘Timing of temperature change relative to the extinction horizon’ would be more clear. Then delete ‘relative’ in next sentence.

Response: Revised. This caption was revised following the suggestion.

Supplementary Materials

In the paragraph titled ‘P4, Wordian and Capitanian (268.8-259.8 Ma)’, perhaps delete the ‘had’: ‘which would have had contributed’

Response: Deleted.

In ‘P5, Wuchiapingian and Changhsingian (259.8-251.9 Ma)’, ‘Armenia’ rather than ‘Armenian’

Response: Revised.

Some additional thoughts that I would be interested in knowing, but I would not require the authors to address if they did not want to, are: Do the slopes in Fig2 A and C (or with warming and cooling times separated) vary systematically with geological age (does the system change over time?) or with paleolatitude of the studied region (does the sampling location systematically bias the results)? A multiple GLS regression would best to ask this question, and the partial

residuals could be plotted so show any significant relationship

Response: We have done a stationarity test for the time series of extinction rates, absolute ΔT , and absolute R . There is no trend for absolute ΔT (Mann-Kendall test, $P = 0.07$) and absolute R (Mann-Kendall test, $P = 0.82$). Extinction rates have a decreasing trend (Mann-Kendall test, $p = 0.01$).

We performed a Pearson-Filon test for the correlation coefficients using R-package cocor. The results show that there is no significant difference between correlation coefficients of extinction rate and ΔT under warming and cooling events (Pearson-Filon's $z = 1.26$, $p = 0.21$). The correlation coefficients of extinction rate and R under warming and cooling events are also not significantly different (Pearson-Filon's $z = 1.34$, $P = 0.18$). A Pearson-Filon test was also performed for the correlation coefficients of different Eras (i.e., Paleozoic, Mesozoic and Cenozoic). The results suggest that there are no significant differences between them. See supplementary Table 3.

To remove the effect of autocorrelation and trend, we used a first differencing method prior to correlation tests. Then, we have performed Pearson-Filon tests on their correlation coefficients again. The results also suggest that there are no significant differences between them, except that the correlation of diff (extinction rates) and diff (ΔT) of the Paleozoic differed from the one of the Mesozoic (Pearson-Filon's $z = 1.97$, $P = 0.048$).

In order to reduce the paleolatitude bias, we only used data from tropical/subtropical regions. To test the latitude bias, we performed Spearman correlation analysis. The results show that there is no significant correlation between paleolatitude and ΔT ($\rho = 0.44$, $P = 0.11$) as well as between paleolatitude and R ($\rho = 0.01$, $P = 0.94$) (see Supplementary Table 2). Paleogeographic map with paleobiological data and paleotemperature data have been added for each Period for additional clarity (see Supplementary Fig. 23).

I am happy for the authors to contact me on carl.reddin@mfn.berlin if they have any questions regarding my review.

Best wishes,
Carl J Reddin

Response: We appreciate your constructive comments, which have been very helpful for improving the quality of this manuscript.

Reviewer #2 (Remarks to the Author):

General comments

This is an interesting manuscript that deals with a very actual topic: the role of temperature changes in past extinction events. The authors correlate two datasets, temperature change and extinction rates, spanning almost the entire Phanerozoic, and find a high, statistically significant, correlation between peaks of extinctions and high rates of temperature change. This results

brings the authors to the conclusion that the main past extinction of the past are linked to a certain threshold of temperature change.

The idea of the study and the conclusions are interesting and the manuscript is, overall, well written. However, I have a few concerns on the way the two datasets are built, and the way analyses are run. These concerns are summarized here, and motivated in more detail below.

Response: We appreciate the positive and constructive comments on our manuscript.

First, paleo-temperature dataset. I really appreciate that the authors built their own dataset by using data derived from the literature. This part of the methods is very well explained (e.g., equations used to get temperature values, how isotopic composition of seawater was estimated, and so on). However, after a look at the supplementary material, I realized that for every time bin, data derive only from one, maximum 2, published papers, from only one region-area, if not locality. This means that the final curve is a biased, regional representation of temperature changes – not a global curve. This should be clearly stated in the methods, or either corrected with the addition of more sites in the dataset.

Response: Thanks. A statement about the data we used for temperature change in time bins was added in the Methods section. The data we used for each time bin were selected from a large paleo-temperature dataset (>20000 oxygen isotopic measurements from the tropical/subtropical regions, see Song et al., 2019) and published literature following the criteria: 1) data with well-constrained ages; 2) data with high time resolution; 3) data from tropical/subtropical regions. Often, this means that there is just a single published dataset that is suitable for analysis. In order to test the reliability of climate events, where possible we added the comparison of temperature data from two different proxies or two different regions. For example, $\delta^{18}\text{O}$ of plankton foraminifera and TEX_{86} were used for the late Eocene, Turonian-Santonian, and Maastrichtian intervals; and conodont $\delta^{18}\text{O}$ from South China and Armenia were used for the Lopingian interval. The results show that data from two different proxies or different regions have a similar magnitude and rate of temperature change (see Supplementary Fig. 22).

Second, fossil dataset. It is not clear at all how the dataset was built. This part of the methods section is very short. Did the authors built their own dataset downloading data from the PBDB? Have they used published data? Have they used a dataset at the species, genus or family level? Does the dataset include data only from marine invertebrates? The authors only state that marine diversity is explored, of what taxa? This is never mentioned in the text. Furthermore, the authors do not explore changes in marine biodiversity, they explore changes in extinction rates, which is very different. Biodiversity (e.g., alpha diversity or richness) is the result of the combination of origination and extinction rates at a certain point in time. This concept should be re-considered throughout the manuscript, as certain sentences are misleading (e.g, line 11 in the abstract).

Response: The fossil data (i.e., GF and 3T extinction rate) we used in the new version of the manuscript are based on our own calculations of data downloaded from the PBDB. For clarity, we added more words in the methods to describe the data: “Gap-filler and three-timer extinction

rates of marine animals were calculated using data from the Paleobiology Database (PBDB, <http://paleobiodb.org>), which was downloaded on 4 January 2021. The fossil dataset includes all metazoans except for Arachnida, Insecta, Ostracoda, and Tetrapoda and consists of 850,840 fossil occurrences of 37,134 genera. These four groups that are excluded (in keeping with some previous studies^{59,65}) were not used because many of them (Arachnida, Insecta, and Tetrapoda) are terrestrial and appear in marine strata. Ostracoda also have a record in terrestrial rocks. Similar to most studies using data from PBDB, we use genus-level occurrences because genus-level taxonomy is better standardized than the taxonomy of species.” (see Lines 340-347).

Biodiversity issue: biodiversity” was replaced by “extinction rate” in the abstract and the main text.

Third, the two datasets have been correlated using Spearman’s rank correlation, the non-parametric version of the Pearson product-moment correlation. However, when working with the analyses of time series, even if with such simple tests, it is necessary to test for autocorrelation between the two datasets. If so, simple transformation are needed (e.g., differentiating the data).

Response: Added. We performed stationarity tests for the time series of extinction rates, absolute ΔT , and absolute R . There is no trend for absolute ΔT (Mann-Kendall test, $p = 0.07$) and absolute R (Mann-Kendall test, $p = 0.82$). Extinction rates have an increasing trend (Mann-Kendall test, $p = 0.01$). To remove the effect of autocorrelation, we used a First Difference Method. The results show that the correlations between Diff extinction rates and Diff ΔT ($r = 0.49$, $p < 0.001$) and Diff R ($r = 0.51$, $p < 0.001$) are still significant.

Besides these three main concerns, which are methodological, I also think that the authors, when discussing their results, should have critically discussed results obtained by similar studies (e.g., Mayhew al. 2012, cited in the introduction) and overall, literature dealing with the interpretation of macroevolutionary patterns and extinctions. Mayhew al. 2012 for instance, clearly state that, even if they found that high extinction rates are correlated with high temperatures, to infer causality is more equivocal. The author mention in the discussion (lines 138-152) that other factors rather than temperature could have caused mass extinction events (volcanic eruptions, bolide impacts and so on), but forget another important point, which come before all of these interpretations, which is related to the structure of the fossil – and geochemical - record. Please see Peters and Foote 2020 (Nature) on the common-cause hypothesis.

Response: Thanks for this constructive suggestion. Literature dealing with the interpretation of macroevolutionary patterns and extinctions has now been cited in our revised discussion, including Mayhew et al., 2012, Peters and Foote, 2002, Stanley and Powell, 2003, and Powell, 2005. We also added more words in the discussion:

“In addition, laboratory experiments show that synergistic stressors of warming and hypoxia are especially lethal to marine animals²⁸.”

“Mayhew et al.³ reported a positive relationship between Phanerozoic temperatures and extinction rates of marine invertebrates. Both intrinsic and extrinsic factors have been used to

explain this correlation ³. Marine ecosystem tended to be dominated by broadly adapted and long-lived taxa during the late Paleozoic ice age, and therefore had sluggish turnover rates ^{29,30}. Warm and productive shelf seas are more conducive to generate high marine diversity, but such settings are vulnerable to hypoxia/anoxia during the warm climate ³⁰.”

“A common cause hypothesis has been used to interpret the correlation between extinction and observed environmental changes, e.g., sea level, both of which are related to the amount of exposed sedimentary rock ^{31,32}. However, the correlation between the extinction rate and temperature change is not an artifact of variability in the amount of exposed rock because the temperature magnitude and rate are independent of variability in the stratigraphic record. Similarly, it could be argued that higher sea levels could lead to a clustering of extinctions in deep water species (e.g. ref. ³²). However, the significant correlations we observe between extinction rates and temperature change are also observed after excluding the deep-water fossil records (Supplementary Table 2).” (see Lines 151-167).”

Said all that, I believe that only once the authors have revised all these points, the manuscript can be considered for publication.

Response: All these points have been revised, and we appreciate the reviewers' positive comments.

Specific points

11. This is misleading. The authors are not studying changes in marine fossil biodiversity, they are studying how extinction rates change through time. This is not marine biodiversity (e.g., number of species through time). Biodiversity depends on the relationship between origination and extinction rates in the same time-interval. So, this sentence is incorrect.

Response: Revised. “biodiversity” is replaced by “extinction rate”.

14. Are the authors dealing with marine animals in general (from invertebrates to vertebrates) or only with marine vertebrates? This should be stated at least one time in the text.

Response: Revised. The fossil dataset includes all metazoans except for Arachnida, Insecta, Ostracoda, and Tetrapoda and consists of 850,840 fossil occurrences of 37,134 genera. (see Lines 341-345). This is now made clear in our revised Methods section.

43-45. As stated below, in the section concerning the methods, it is not clear what kind of data have been analysed (species, genus, family level?).

Response: Added. We analysed the extinction rate at genus level (see Lines 346-347).

78-79. As stated above, high extinction rates do not necessarily imply a biodiversity loss, if origination rates are high. This sentence should be reworded.

Response: Reworded. “biodiversity loss” was replaced by “extinction rate”.

90-92. The differences between the two methods used to measure extinction rates should be better explained.

Response: Added. More words were added in the Methods to address the differences among the GF, 3T, and boundary crosser methods: “Both the GF and 3T rates are calculated from occurrence data and have a higher accuracy than the boundary-crosser method ⁵⁹. The boundary-crosser method is more susceptible to major biases such as the Signor-Lipps effect and sampling bias because it use full age-ranges instead of investigating sampling patterns across limited temporal windows ⁵⁹. The 3T method can be noisy in the case of high turnover rates and poor sampling. The GF method is more precise when sampling is very poor.” (see Lines 334-339).

103-107. How did you define which the minor extinction events are? Which is the reference publication? Some authors could argue that some of the extinctions that you point out here, e.g, the Cenomanian-Turonian, are not true extinctions but a stratigraphic artefact (e.g. Gale et al. 2000 Journal of the Geological Society of London; Smith et al. 2001 Paleobiology). The reason why I am saying this is that, for instance, the Cenomanian-Turonian extinction, like others (e.g., early Toarcian, late Ordovician), are related to major sea level rises, which would cluster last occurrences of deep water species, making the rate of extinction higher. Such transgressions are presumably related to warming events, and it becomes than difficult to understand what is the main cause of the extinction (please see Holland 2010 Palaeontology and Peters and Foote 2002 Nature: the common case hypothesis).

Response: Thanks. The definition of minor extinctions mainly refers to Hallam and Wignall 1997 and published literature listed in the Supplementary Table 4. For the common cause hypothesis, we added a new paragraph to discuss this issue “A common cause hypothesis has been used to interpret the correlation between extinction and observed environmental changes, e.g., sea level, both of which are related to the amount of exposed sedimentary rock ^{31,32}. However, the correlation between extinction rate and temperature change is not an artifact of variability in the amount of exposed rock because the temperature magnitude and rate are independent of variability in the stratigraphic record. Similarly, it could be argued that higher sea levels could lead to a clustering of extinctions in deep water species (e.g. ref. ³²). However, the significant correlations we observe between extinction rates and temperature change are also observed after excluding the deep-water fossil records (Supplementary Table 2).” (see Lines 159-167).

We agree with the reviewer that climate-related sea level change would potentially cluster last occurrences of deep water species. To investigate this, we repeated our analyses using only shallow-water species to analyze the relationship between temperature change and extinction rate. The results show significant correlations between both magnitude and rate of temperature change and extinction rate of shallow-water taxa (see Supplementary Table 2), supporting our conclusion.

130-137. The examples reported here refer to laboratory experiments which cannot be compared with at which temperature changed through extinction events. I would rather support the results

with other examples related, for instance, to the impossibility of species to track their preferred habitat (e.g., Sunday et al. 2012 Nature Climate Change).

Response: Thanks. The laboratory experiments are important for us to understand the relationship between temperature change and biotic response, which provides the theoretical basis for the explanation of phenomenon in geologic records. We choose to keep these sentences. Nevertheless, we have added the example (Sunday et al. 2012 Nature Climate Change) following the reviewer's suggestion (see Lines 132-133).

Figure 2. In a and c, are you using the absolute values of ΔT and R? If so, explain it in the caption. Also double check colors (they do not match between figure and caption).

Response: Revised. "absolute" has been added before " ΔT and R". Colors have been checked and revised.

Methods

Paleotemperature estimators: It has been a very good idea to select the studied sample by paleo-latitude, including only data coming from tropical and subtropical regions. What is lacking, and would be added, is the range of environments from which the data come from: shallow subtidal, deep subtidal (below or above storm wave base), shallow carbonate reef? I believe that this information should be added.

Response: Thanks. The environmental information of data have been added in the Table S1, and we now provide maps of our data to clearly show where the data are from paleogeographically.

Temperature Database & Supplementary Dataset: I had a quick look at the dataset, and have a couple of concerns. First, the dataset is not complete, some time intervals are not listed. Missing time intervals are: T1, J4, J5, J6, K1, K2, k3, k8, Pg2, Pg3, Pg4. Second, from most of the time intervals, data come only from 1 publication and/or 1 locality. If we focus on the Jurassic, for instance, in J1, data come only from ref. Korte and Heselbo 2011, which are all sections from the same locality in UK; for J2-J3 data come only from refs. Suan et al. 2008, 2010, which are all sections from Portugal. However, there are other published papers with published stable isotope oxygen data from shells from the early Jurassic from other localities in the Palaeotethys, which have not been included, why? (e.g., Saelen et al. 1996, Palaios, Mc Arthur et al. 2000, Dera et al. 2009 EPSL, Danise et al. 2019 Palaeo 3, Ulmann et al 2020 Scientific Reports). I supposed that this would be the case also for other time-intervals.

Response: We used recalculated moving average curves for most time bins, and these data and curves are listed in Supplementary Figs. 1-17. For other time bins, we directly extracted temperature data from previously published and well-known smoothed paleotemperature curves. This was addressed and cited in the Supplementary Methods (see 'Temperature database' section). For the Early Triassic time bin (T1), we used the well-known smoothed SST curve established using conodont oxygen isotope data from South China (Sun et al., 2012). For the Middle and Late Jurassic bins (J4 and J5), we used the well-known paleotemperature curve

established by using belemnite oxygen isotope data from Europe (Dera et al., 2011). This smoothed curve was generated using Kernel regression with a bandwidth of 0.5 Myr (Dera et al. 2011). For J6, we used the smoothed temperature curve established using oyster oxygen isotope data from France (Brigaud et al., 2008). For K1 and K2, we used the smoothed temperature curve established using belemnite oxygen isotope data from France (Bodin et al., 2009). For K3, we used a moving average curve established based on TEX₈₆ data from DSDP site 398 (Naafs et al., 2016). For K8, we used the average magnitude of temperature change based on TEX₈₆ data from three sites in eastern USA (Vellekoop et al., 2014, 2016). For the Eocene bins (Pg2-Pg4), we used the smoothed paleotemperature curve established based on multiple proxies including TEX₈₆, $\delta^{18}\text{O}$ and Mg/Ca of planktonic foraminifer (Cramwinckel et al., 2018). The Eocene SST curve was generated using local regression model with a bandwidth of <0.5 Myr (Cramwinckel et al., 2018). For clarity, the statement of data we used for temperature change in time bins was added in the methods (see Lines 226-239).

The data we used for the 45 time bins were selected from a large paleo-temperature dataset, i.e., >20000 oxygen isotopic measurements from the tropical/subtropical regions (Song et al., 2019) as well as published literature based on other temperature proxies by adopting the following criteria: 1) data with well-constrained ages; 2) data with high time resolution; 3) data from tropical/subtropical regions. There are often several paleotemperature publications for a time bin. However, when we focus on the interval with most obvious temperature change in a given time bin and consider the above selection criteria, there are only one/two publications left. Take the T5 (Rhaetian) bin for example, at least six papers have reported Rhaetian paleotemperature data, i.e., Veizer et al., 1999, CG; Korte et al., 2005, Paleo3; Korte et al., 2009, J. Geol. Soc.; Mette et al., 2012, Paleo3; Trotter et al., 2015, EPSL; Korte et al., 2017, N. Jb. Geol. Palaeont. Abh. However, only one (Korte et al., 2009) covers the latest Rhaetian interval, during which the most significant temperature change happened, coincident with the end-Triassic mass extinction. The other five papers only cover the early and middle Rhaetian intervals. This is also the case for other time bins.

In order to test the reliability of climate events, where possible we added a comparison of temperature data from two different proxies and/or two different regions. For example, $\delta^{18}\text{O}$ of planktonic foraminifera and TEX₈₆ were used for the late Eocene, Turonian-Santonian, and Maastrichtian intervals; and conodont $\delta^{18}\text{O}$ from South China and Armenia were used for the Lopingian interval. The results show that data from two different proxies or different regions have a similar magnitude and rate of temperature change (see Supplementary Fig. 22).

Extinction rates: This is the most deficient part of the methods. The authors use two methods to estimate of extinction rates, the gap-filler (GF) and three-timer (3T), used for the first time by Alroy 2014. I do not understand if they made their own estimates downloading data from the PBDB database, or if they used the estimates of Alroy 2014. In the first case, it is not stated if they are using data at the species, genera, or family level, and what higher taxonomic levels they have included in the datasets (only marine invertebrates? Marine vertebrates too?). Moreover, have they included marine taxa from all the marine environments or excluded, for instance, deep water taxa that were potentially less affected by SST changes? In the second instance, if they

used the data from Alroy, unfortunately they are using a dataset that needs to be updated, as the PBDB is continuously updated with new data and mistakes are continuously corrected. Moreover, the authors should state why they decided to use these 2 estimates of extinction rates (for instance, Alroy in his 2014 paper use 3 different estimates).

Response: Added.

The fossil data (i.e., GF and 3T extinction rate) we used in the new version of the manuscript are our own dataset that we downloaded from the PBDB. We then calculated extinction rates using these raw data. Our Methods section has been revised to make everything more clear. Notably, we added more words in the Methods to describe the data: “Gap-filler and three-timer extinction rates of marine animals were calculated using data from the Paleobiology Database (PBDB, <http://paleobiodb.org>), which was downloaded on 4 January 2021. The fossil dataset includes all metazoans except for Arachnida, Insecta, Ostracoda, and Tetrapoda and consists of 850,840 fossil occurrences of 37,134 genera. These four groups are excluded (in keeping with some previous studies (Alroy et al., 2008, 2014)) because many of them (Arachnida, Insecta, and Tetrapoda) are terrestrial and appear in marine strata. Ostracoda also has a record in terrestrial rocks. Similar to most studies using data from PBDB, we use genus-level occurrences because genus-level taxonomy is better standardized than the taxonomy of species.” (see Lines 340-347).

We also analyzed extinction rates of marine taxa from just shallow-water facies (i.e. deep water species were removed). The results show that both GF and 3T extinction rates show significant correlations with the magnitude and rate of temperature change (see Supplementary Table 2), supporting our conclusions. The results have been added in the manuscript (see Lines 165-167).

We added a statement about selecting the two estimates of extinction rates in Lines 333-339: “There are a few methods to calculate extinction rate in geological history, e.g., boundary-crosser, gap-filler (GF) and three-timer (3T) rates^{59,64,65}. Both the GF and 3T rates are calculated from occurrence data and have a higher accuracy than the boundary-crosser method⁵⁹. The boundary-crosser method is more susceptible to major biases such as the Signor-Lipps effect and sampling bias because it use full age-ranges instead of investigating sampling patterns across limited temporal windows⁵⁹. The 3T method can be noisy in the case of high turnover rates and poor sampling. The GF method is more precise when sampling is very poor.”

It would be very useful, for each bin (or at least for each Period), to have a map showing from which locations the paleobiological data come from, and from which location the paleotemperature data come from. This would help to understanding the structure of the dataset, and would highlight possible biases.

Response: Added. Paleogeographic maps with paleobiological data and paleotemperature data plotted have been added for each Period (see Supplementary Fig. 23).

Reviewer #3 (Remarks to the Author):

Although the premise of the MS is very interesting, timely and of broad interest, at present I find some of the methods to be inappropriate and lacking adequate description in both the main body text and in the methods.

Response: We appreciate the comments on our manuscript and have now addressed all of the issues raised.

Three key issues:

1. The extinction metrics are not well described in the main body text, there is more detailed info in the methods, but a simple explanation of what each method is trying to do would really help the reader with the flow of the MS.

Response: Thanks. A detailed description of extinction metrics has been added in the Methods: "There are a few methods to calculate extinction rate in geologic time, e.g., boundary-crosser, gap-filler (GF) and three-timer (3T) rates^{59,64,65}. Both the GF and 3T rates are calculated from occurrence data and have a higher accuracy than the boundary-crosser method⁵⁹. The boundary-crosser method is more susceptible to major biases such as the Signor-Lipps effect and sampling bias because it use full age-ranges instead of investigating sampling patterns across limited temporal windows⁵⁹. The 3T method can be noisy in the case of high turnover rates and poor sampling. The GF method is more precise when sampling is very poor.

Gap-filler and three-timer extinction rates of marine animals were calculated using data from the Paleobiology Database (PBDB, <http://paleobiodb.org>), which was downloaded on 4 January 2021. The fossil dataset includes all metazoans except for Arachnida, Insecta, Ostracoda, and Tetrapoda and consists of 850,840 fossil occurrences of 37,134 genera. These four groups are excluded (in keeping with some previous studies^{59,65}) because many of them (Arachnida, Insecta, and Tetrapoda) are terrestrial and appear in marine strata. Ostracoda also has a record in terrestrial rocks. Similar to most studies using data from PBDB, we use genus-level occurrences because genus-level taxonomy is better standardized than the taxonomy of species." (see Lines 333-347).

2. The use of "uniform calibrations" to proxy data. This is one of my biggest concerns. The reason that many different T calibration methods are applied across this vast literature is because it is necessary. By using only one oxygen isotope T equation, for example, much of the complexity of the sources and biases on these data are overlooked. Also, the choice of d18O eqn seems odd. Why this paleo temp equation? The calibration range of this is not sufficient to accommodate very warm worlds, such as the Cretaceous, and the calibration of d18O is particularly non-linear towards higher temps e.g. 30 degC +. Surely something like Kim and O'Neill with inorganic calcite over a very wide T range would have been a better choice for this? I am not an expert in all of the T proxies, so I cannot comment on whether this "uniform calibration" approach is suitable for all of the proxies listed, but do not think it is suitable for d18O even when you are just trying to capture magnitudes of change rather than explicit values.

Response: Although there are limitations in applying a single d18O proxy, it is not clear what the best equations are for different time intervals. As such, we think that our simpler approach

removes the possibility of bias that could arise by using different equations for different time intervals.

For the calcite d18O equation, we previously used the equation from O'Neil et al. 1969 reformulated by Hays and Grossman 1991. The calibration range of this equation is quite wide: 0-60°C, which can be applied for all the periods in the Phanerozoic. Following the reviewer's suggestion, we updated this equation for calcite d18O by using the equation of Kim and O'Neil 1997 (see Lines 252-253).

3. My final main concern is with the age control. It is central to the approach, which is attempting to tie T change with extinction events and yet the methodology is very brief, to the extent that I would not be able to reproduce what they did based on the information given. "In our database, only one warming event, at the Permian-Triassic boundary, meets this criterion. Other dates were obtained using a comprehensive approach including isotope geochronology, astrochronology, and biostratigraphy with reference to the Geologic Time Scale 2012" How were these used/integrated? Was there a rank order of what was the most reliable? What if the different archives didn't give the same dates, how did you reconcile this? Were there any conflicting archives? Dating is non-trivial, and this seems to only get a cursory mention? The latter two points must be better explained and justified for this current approach to stand up to further scrutiny.

Response: Thanks. A more detailed description of ages has now been added in the Methods (see Lines 284-289). Age data were applied in the following order of priority: isotope geochronologic age, astrochronologic age, and biostratigraphic age. If isotope geochronologic ages were available, we gave priority to these absolute dates. In the absence of absolute age data, astrochronologic ages were preferred. If neither of these numerical data was available, the climatic events were timed based on the age of biozones in the Geologic Time Scale 2012. The relative age within the same biozone was constrained based on the stratigraphic position.

Other minor comments on the MS.

Line 43: Hmmm this is interesting as much of climate change is amplified at the poles and it seems odd to focus on low lats, which its likely high lat communities were likely more impacted. Is this because of the availability of info in the database, if so I would state this, otherwise it seems a bit shortsighted.

Response: Yes, the paleotemperature data are primarily from low latitudes. We have now added paleogeographic maps to show the positions of our paleotemperature data (see Supplementary Fig. 23).

Lines44-45: these terms need to be introduced. What are they and how exactly are they measuring extinction rate? If you do not know what these are it makes the rest of the discuss rather opaque.

Response: Added. More words were added in the Methods to introduce the GF, 3T, and boundary crosser methods as well as the fossil database: "Both the GF and 3T rates are calculated from

occurrence data and have a higher accuracy than the boundary-crosser method ⁵⁹. The boundary-crosser method is more susceptible to major biases such as the Signor-Lipps effect and sampling bias because it use full age-ranges instead of investigating sampling patterns across limited temporal windows ⁵⁹. The 3T method can be noisy in the case of high turnover rates and poor sampling. The GF method is more precise when sampling is very poor.” (see Lines 334-339); “The fossil dataset includes all metazoans except for Arachnida, Insecta, Ostracoda, and Tetrapoda and consists of 850,840 fossil occurrences of 37,134 genera. These four groups that are excluded (in keeping with some previous studies ^{59,65}) were not used because many of them (Arachnida, Insecta, and Tetrapoda) are terrestrial and appear in marine strata. Ostracoda also have a record in terrestrial rocks. Similar to most studies using data from PBDB, we use genus-level occurrences because genus-level taxonomy is better standardized than the taxonomy of species.” (see Lines 341-347).

Lines 193-194: this is rarely presented for Cenozoic data and so you are only listing one avenue of filtering. what others? an "e.g." in this regard is not good enough, it needs to be the full list

Response: Added. The full list was added here: “i.e., $\delta^{18}\text{O}$ data from localities with abnormal salinities and upwelling systems, $\delta^{18}\text{O}$ data from carbonate fossil shells with Mn > 250 ppm and Sr < 400 ppm, TEX_{86} data with BIT > 0.4” (see Lines 212-213).

Reviewer comments, second round –

Reviewer #1 (Remarks to the Author):

I congratulate the authors on a very nice manuscript that I hope to see published soon, as I expect it will stimulate much discussion in the field. The authors have addressed my comments to my satisfaction. I add just a couple of notes the authors may wish to address:

L59: I would have made the results based on differenced time series the main results. Otherwise, I would add a note up front that the effect of temporal autocorrelation was also assessed and didn't change the main results, as many readers will be wondering by this point. A note could also be added to figure 2 caption.

L71: This correlation is insignificant but is not mentioned as such.

L337: "it uses"

L357: "Autocorrelation test". There is actually no autocorrelation test (e.g. Moran's I) described here. Perhaps then instead call this section something like, "Autocorrelation and stationarity"

Fig. 3 caption: Perhaps add a note here that magnitude in the sense of this paper only pertains to rapid "within-interval" magnitude e.g. the high magnitude warming following the Permo-Carboniferous glaciation was very slow (was not "within-interval") and therefore not selected by this study

Thanks for a very interesting read and best wishes,
Carl J Reddin

Reviewer #2 (Remarks to the Author):

The authors have made a very good effort in addressing the points highlighted during the first round of review.

I believe that this effort makes the manuscript suitable for publication.

Reviewer #4 (Remarks to the Author):

In this paper, Song et al. collect temperature data throughout the Phanerozoic to compare the rate of T changes to the rate of extinction. They find a significant relationship and further suggest that the threshold for mass extinction is 5.5C of warming at a rate of at least 10C/Myr.

I think this is an elegant paper and I really enjoyed reading it. I commend the authors for doing such a comprehensive analysis. However, in order to make the conclusions of the paper more robust, the analysis needs to have more statistical rigor in terms of quantifying uncertainties. Both the temperature and age estimates carry uncertainties, and these need to be propagated into the calculations of rate of change, such that all estimates of ΔT , R, and extinction rate have confidence intervals associated with them. This will assist with hypothesis testing (correlation calculations, etc) and comparison to the rate of modern warming.

First, let's talk proxies and calibrations.

1) For d_{18O} , KimONeil/Bemis and Lecuyer are fine, but they need to have uncertainties associated with them that are somewhat reasonable for the biological systems in question. The minimum uncertainty is analytical precision, and the "real" uncertainty is certainly bigger than that. I can share our approach for the deepMIP analysis of d_{18O} of forams, which I think could be used here (Hollis et al., 2019, GMD, <https://doi.org/10.5194/gmd-12-3149-2019>). In that paper, we chose

to use the Bemis high light and low light equations, then calculate temperatures by bootstrapping between them. You also might want to attach an uncertainty in the seawater d18O determination, since this can vary spatially, and bootstrap over that. In short, for d18O, you can propagate some amount of both proxy and seawater uncertainty simply through Monte Carlo simulation. Alternatively, you could use the "all" option in our BAYFOX Bayesian model for d18O in forams (<https://github.com/jesstierney/bayfoxm>), which should be applicable to brachiopods, oysters, and belemnites since like forams they fall close to the theoretical slope. This will give you an uncertainty on T comparable to that associated with d18O of forams in coretops in the present day and you won't have to make an arbitrary decision about that. You will have to make a decision about seawater uncertainty still.

There is one more factor to consider as well: pH. We did not correct our d18O estimates in deepMIP for pH, but we should have done so. For context, see Zeebe, 2001, P3 ([https://doi.org/10.1016/S0031-0182\(01\)00226-7](https://doi.org/10.1016/S0031-0182(01)00226-7)). During global warming/extinction events that are related to CO₂, ocean pH is going to be dropping, which means that the pH effect will impact your calculation of rates of change.

Screening d18O: the paper mentions that the d18O data were screened for diagnosis, but I'd like some more details. There are "obvious" cases on diagenesis, such as, for instance, d18O values that are unrealistically negative. But, for the marine planktic data in particular, the diagenesis generally works in the other way (d18O values are too enriched) and it can be subtle. Some have argued that only "glassy" foram data should be used for this reason. What criteria did the authors use? Even though this paper is only interested in relative change, diagenetic overprinting would attenuate the magnitude of change across warming events, so it is important to consider.

2) TEX86: Kim et al 2010 TEXH equation is not an acceptable calibration. The problem is that it was calculated incorrectly, such that it suffers from regression dilution. This is discussed in the original BAYSPAR paper (Tierney & Tingley, 2014, GCA) but also in Hollis et al., 2019. The practical outcome of this is that Kim10 underpredicts warm SSTs and overpredicts cold SSTs, such that you will get a bias towards a smaller range of change from it. This is a statistical flaw. For this reason, you have to use BAYSPAR (<https://github.com/jesstierney/BAYSPAR>). For deep time, you'll need to use the analog mode. Watch the threshold setting, and make sure it includes enough analogs; this can be a problem for the really high TEX86 values in the Cretaceous for example. The output from BAYSPAR is a very generous estimate of uncertainty. Since you are only interested in relative changes, you might want to remove the uncertainty associated with the absolute values. I usually do this by sorting the posterior ensemble from low to high then subtracting off the first value or the mean. I then add "downcore" uncertainty back as Normal random error N(0,0.5) (1-sigma uncertainty of half a degree). This isn't totally satisfactory because it's a posthoc split of the error variance, but it is the best solution I have right now. This procedure is described in our recent Nature paper on the LGM (<https://doi.org/10.1038/s41586-020-2617-x>). If you go the BAYFOX route for d18O, I would use this same approach as well.

-Screening of TEX86: BIT is actually not a great predictor of "problematic" TEX distributions. I would instead - where possible - use the methane Index (Zhang et al. 2011, EPSL, <https://doi.org/10.1016/j.epsl.2011.05.031>) and delta-ring index (Zhang et al., 2015, <https://doi.org/10.1002/2015PA002848>). While both papers give some guidelines, there is not clear cutoffs for these. From practical experience, I can tell you that high methane index is particularly good at singling out problem distributions so I tend to use that one the most. These indices require that the authors report relative abundances of GDGTs, which many of the earlier Cretaceous studies do not have, so in that case yeah, you will be stuck with BIT.

3) Mg/Ca: Mg/Ca is a bit of nightmare because it is sensitive to salinity, pH, the saturation state (omega), the lab cleaning method, and on the timescales relevant to this paper, the Mg/Ca of the seawater. You have to make some assumptions about these to get accurate estimates. Anand et al is not going to cut it. Fortunately I've done the hard work for you - you can use BAYMAG (<https://github.com/jesstierney/BAYMAG>). The paper is here (<https://doi.org/10.1029/2019PA003744>). It has a seawater curve built in, so you can correct for Mg/Ca seawater evolution. You will need to make some choices for pH and omega. Omega is pretty important, but unfortunately it is usually impossible to get an actual paleoestimate for it.

For your study, I might suggest just holding it at a value of 5, which basically assumes that it doesn't have much of an effect. The cleaning method should be reported in the original study. Salinity is a pretty minor effect, so holding it constant is reasonable unless you think some big changes occurred.

OK, now the age uncertainties. As far as I can tell, there isn't a consideration of this, but there needs to be. Based on the available chronological constraints for each event, one should be able to estimate the uncertainty simply by Monte Carlo-ing different possible age models that fall within the acceptable range. In "shallow-time" work I use BACON software (rbacon package in R) to do this with some Bayesian constraints, but even a simple MC version would be sufficient here. What you want to achieve here is an estimate of the uncertainty in time, which translates to the uncertainty in the rate of change of both temperature and extinction.

In the end, you will have an ensemble of possible SST values from the proxies, and an ensemble of possible age models. Bootstrap sample across both to get a "final" estimate of uncertainty on your rates. Use this ensemble method to add error bars to all the figures, and for hypothesis testing.

Beyond the need for uncertainty quantification, my only other content suggestion concerns the K/Pg extinction, which is unique as it is associated with an impact. Hull et al 2020 (Science) showed that the volcanic outgassing and associated global warming near the K/Pg boundary predates the extinction and found no evidence that the extinction is related to the late Maastrichtian warming. The cooling that follows this warming, moving into the K/Pg, is relatively slow and small-magnitude, hence the authors conclude that it is the impact itself and associated climate changes that cause the extinction. Given the coarse resolution of the analysis in this paper, it is not clear to me whether the correlation analysis is reflecting the late Maastrichtian warming/cooling, or the volcanic winter that is associated with the impact. From a cause-and-effect point of view, the analysis of the link b/t T and extinction for the K/Pg should be done specifically on the volcanic winter since that is the actual climate change associated with extinction.

One more statistical note on the correlations: Reviewer 1 already brought this up, but the p-values on all correlations should account for the reduced degrees of freedom that comes from the autocorrelation in the time series plotted in Fig. 1. There are several methods to do this, my personal preference is to use the nonparametric method of Ebisuzaki et al. 1997 J. Climate ([https://doi.org/10.1175/1520-0442\(1997\)010<2147:AMTETS>2.0.CO;2](https://doi.org/10.1175/1520-0442(1997)010<2147:AMTETS>2.0.CO;2)). I see that the authors addressed this with first differences, but that doesn't always do the job, so I recommend checking with Ebisuzaki to be sure.

Finally, regarding the T compilations, I agree with Reviewer 2 that it is too limited. In particular, the Cenozoic events are sparsely represented and there is no need for that because there are tons of data out there. For the Paleogene, the authors might benefit from using the deepMIP compilation for the PETM and the Eocene (Hollis et al 2019) which is more comprehensive than what is used here. There also seems to only be one site for the Neogene time intervals, but I suspect there is much more out there; there should be some UK37 data available for example (?). I think it is important to use as much data as possible, so that the analysis isn't biased by use of a single or only a few sites. I don't see that the authors have addressed this comment by Reviewer 2 head-on yet, but I think that they should.

Here are a few line-by-line comments:

- Line 194: Perhaps update this discussion to SSP5-8.5 to reflect the AR6 scenarios.
- Line 208: belemnite Mg/Ca
- Line 214: BIT is probably not the best criterion, see above comments on treatment of TEX86.
- Line 226: Moving averages have the undesirable effect of "time-shifting" your data so I wouldn't use them. Use a Gaussian smooth instead if you would like to minimize sensitivity to outliers. Most statistics software (Matlab, R) have built-in functions for this. Apply the same method to all of your sites, rather than variously relying on moving averages, kernel regression, etc.

Good luck with the revisions and feel free to contact me if anything in the above is unclear.

-Dr. Jessica Tierney

Note: original reviewer comments in black; responses of authors in blue

REVIEWER COMMENTS

Reviewer #1 (Remarks to the Author):

I congratulate the authors on a very nice manuscript that I hope to see published soon, as I expect it will stimulate much discussion in the field. The authors have addressed my comments to my satisfaction. I add just a couple of notes the authors may wish to address:

Response: We appreciate the reviewer's very positive comments.

L59: I would have made the results based on differenced time series the main results. Otherwise, I would add a note up front that the effect of temporal autocorrelation was also assessed and didn't change the main results, as many readers will be wondering by this point. A note could also be added to figure 2 caption.

Response: Added. The requested notes have now been added in the main text and the caption of figure 2.

L71: This correlation is insignificant but is not mentioned as such.

Response: Revised. "insignificant" has been added to this sentence.

L337: "it uses"

Response: Revised.

L357: There is actually no autocorrelation test (e.g. Moran's I) described here. Perhaps then instead call this section something like, "Autocorrelation and stationarity"

Response: Revised. "Autocorrelation test" has been replaced by "Autocorrelation analysis".

Fig. 3 caption: Perhaps add a note here that magnitude in the sense of this paper only pertains to rapid "within-interval" magnitude e.g. the high magnitude warming following the Permo-Carboniferous glaciation was very slow (was not "within-interval") and therefore not selected by this study

Response: Added. The requested text has been added in the caption of figure 3, i.e., "the plotted magnitudes and rates pertain to those that occur within each of the 45 defined time intervals."

Thanks for a very interesting read and best wishes,

Carl J Reddin

Response: Thank you for your useful suggestions.

Reviewer #2 (Remarks to the Author):

The authors have made a very good effort in addressing the points highlighted during the first round of review.

I believe that this effort makes the manuscript suitable for publication.

Response: We appreciate the reviewer's positive comments.

Reviewer #4 (Remarks to the Author):

In this paper, Song et al. collect temperature data throughout the Phanerozoic to compare the rate of T changes to the rate of extinction. They find a significant relationship and further suggest that the threshold for mass extinction is 5.5C of warming at a rate of at least 10C/Myr.

I think this is an elegant paper and I really enjoyed reading it. I commend the authors for doing such a comprehensive analysis. However, in order to make the conclusions of the paper more robust, the analysis needs to have more statistical rigor in terms of quantifying uncertainties. Both the temperature and age estimates carry uncertainties, and these need to be propagated into the calculations of rate of change, such that all estimates of deltaT, R, and extinction rate have confidence intervals associated with them. This will assist with hypothesis testing (correlation calculations, etc) and comparison to the rate of modern warming.

Response: We appreciate the positive and constructive comments on our manuscript.

First, let's talk proxies and calibrations.

1) For d18O, KimONeil/Bemis and Lecuyer are fine, but they need to have uncertainties associated with them that are somewhat reasonable for the biological systems in question. The minimum uncertainty is analytical precision, and the "real" uncertainty is certainly bigger than that. I can share our approach for the deepMIP analysis of d18O of forams, which I think could be used here (Hollis et al., 2019, GMD, <https://doi.org/10.5194/gmd-12-3149-2019>). In that paper, we chose to use the Bemis high light and low light equations, then calculate temperatures by bootstrapping between them. You also might want to attach an uncertainty in the seawater d18O determination, since this can vary spatially, and bootstrap over that. In short, for d18O, you can propagate some amount of both proxy and seawater uncertainty simply through Monte Carlo simulation. Alternatively, you could use the "all" option in our BAYFOX Bayesian model for d18O in forams (<https://github.com/jesstiemey/bayfoxm>), which should be applicable to brachiopods, oysters, and belemnites since like forams they fall close to the theoretical slope. This will give you an uncertainty on T comparable to that

associated with d18O of forams in coretops in the present day and you won't have to make an arbitrary decision about that. You will have to make a decision about seawater uncertainty still.

Response: Thanks for this helpful advice on handling uncertainties. In our revised manuscript, we have used the pooled BAYFOX Bayesian model (Malevich et al., 2019) for d18O temperature estimations in planktonic forams as well as other carbonate fossils, i.e., brachiopods, oysters, and belemnites. To do this, we used the R package "bayfoxr". For phosphate fossil conodont data, we used Monte Carlo simulations to propagate parameter uncertainties in temperature estimation, based on equation (2) of Pucéat et al., 2010.

There is one more factor to consider as well: pH. We did not correct our d18O estimates in deepMIP for pH, but we should have done so. For context, see Zeebe, 2001, P3 ([https://doi.org/10.1016/S0031-0182\(01\)00226-7](https://doi.org/10.1016/S0031-0182(01)00226-7)). During global warming/extinction events that are related to CO₂, ocean pH is going to be dropping, which means that the pH effect will impact your calculation of rates of change.

Response: Thanks for this suggestion. We agree that seawater pH is an important factor, and we thank the reviewer for drawing this to our attention. Unfortunately, pH estimates for the Phanerozoic time bins are few, and their values are highly uncertain (Halevy and Bachan, 2017, Science). Therefore, we did not correct d18O estimates in this manuscript. However, we have now added a discussion of the potential effects of pH drops on the magnitude of climate change (see lines 241-245).

Screening d18O: the paper mentions that the d18O data were screened for diagenesis, but I'd like some more details. There are "obvious" cases on diagenesis, such as, for instance, d18O values that are unrealistically negative. But, for the marine planktic data in particular, the diagenesis generally works in the other way (d18O values are too enriched) and it can be subtle. Some have argued that only "glassy" foram data should be used for this reason. What criteria did the authors use? Even though this paper is only interested in relative change, diagenetic overprinting would attenuate the magnitude of change across warming events, so it is important to consider.

Response: We have added this requested information. Detailed methods of screening d18O have been added in the revised manuscript, i.e., "Oxygen isotope values that were likely affected by diagenetic alteration or local effects were removed from the database, i.e., $\delta^{18}\text{O}$ data from localities with abnormal salinities and upwelling systems. Other screening criteria include $\delta^{18}\text{O}$ values that are unrealistically negative or positive and $\delta^{18}\text{O}$ values from carbonate fossil shells with Mn > 250 ppm and Sr < 400 ppm. In addition, $\delta^{18}\text{O}$ records that show offsets for a given interval between different sites were likely influenced by diagenesis." (see lines 245-250). For most Cenozoic time bins, we used TEX86 temperature estimators. We used both planktic d18O data (three sites) and TEX86 (two sites) for the Pg4 bin and they show similar results in both trends and magnitudes of temperature change, suggesting that these d18O data are not altered by diagenesis.

2) TEX86: Kim et al 2010 TEXH equation is not an acceptable calibration. The problem is that it was calculated incorrectly, such that it suffers from regression dilution. This is discussed in the original BAYSPAR paper (Tierney & Tingley, 2014, GCA) but also in Hollis et al., 2019. The practical outcome of this is that Kim10 underpredicts warm SSTs and overpredicts cold SSTs, such that you will get a bias towards a smaller range of change from it. This is a statistical flaw. For this reason, you have to use BAYSPAR (<https://github.com/jesstierney/BAYSPAR>). For deep time, you'll need to use the analog mode. Watch the threshold setting, and make sure it includes enough analogs; this can be a problem for the really high TEX86 values in the Cretaceous for example. The output from BAYSPAR is a very generous estimate of uncertainty. Since you are only interested in relative changes, you might want to remove the uncertainty associated with the absolute values. I usually do this by sorting the posterior ensemble from low to high then subtracting off the first value or the mean. I then add "downcore" uncertainty back as Normal random error $N(0,0.5)$ (1-sigma uncertainty of half a degree). This isn't totally satisfactory because it's a posthoc split of the error variance, but it is the best solution I have right now. This procedure is described in our recent Nature paper on the LGM (<https://doi.org/10.1038/s41586-020-2617-x>). If you go the BAYFOX route for d18O, I would use this same approach as well.

Response: We thank the reviewer for this helpful advice. We have used the analog mode of the matlab package BAYSPAR (Tierney and Tingley, 2014; 2015) for TEX86 data to estimate temperature. The prior was set using Kim's equation with a large standard deviation.

-Screening of TEX86: BIT is actually not a great predictor of "problematic" TEX distributions. I would instead - where possible - use the methane Index (Zhang et al. 2011, EPSL. <https://doi.org/10.1016/j.epsl.2011.05.031>) and delta-ring index (Zhang et al., 2015, <https://doi.org/10.1002/2015PA002848>). While both papers give some guidelines, there is not clear cutoffs for these. From practical experience, I can tell you that high methane index is particularly good at singling out problem distributions so I tend to use that one the most. These indices require that the authors report relative abundances of GDGTs, which many of the earlier Cretaceous studies do not have, so in that case yeah, you will be stuck with BIT.

Response: Thanks for this suggestion. In the revised manuscript, Methane Index (MI >0.5, Zhang et al., 2011), delta-Ring Index ($\Delta RI > |0.3|$, Zhang et al., 2016), %GDGT-0 (>67%, Sinninghe Damste et al., 2012), fCren':Cren' + Cren (>0.25, O'Brien et al., 2017), and BIT (>0.4, Inglis et al., 2015) have been used to predict "problematic" TEX86 distributions.

3) Mg/Ca: Mg/Ca is a bit of nightmare because it is sensitive to salinity, pH, the saturation state (omega), the lab cleaning method, and on the timescales relevant to this paper, the Mg/Ca of the seawater. You have to make some assumptions about these to get accurate

estimates. Anand et al is not going to cut it. Fortunately I've done the hard work for you - you can use BAYMAG (<https://github.com/jesstierney/BAYMAG>). The paper is here (<https://doi.org/10.1029/2019PA003744>). It has a seawater curve built in, so you can correct for Mg/Ca seawater evolution. You will need to make some choices for pH and omega. Omega is pretty important, but unfortunately it is usually impossible to get an actual paleoestimate for it. For your study, I might suggest just holding it at a value of 5, which basically assumes that it doesn't have much of an effect. The cleaning method should be reported in the original study. Salinity is a pretty minor effect, so holding it constant is reasonable unless you think some big changes occurred.

Response: Thanks for this very helpful advice on dealing with Mg/Ca data, which we agree can be problematic! The Mg/Ca data that we can use in our study (e.g., with sufficiently high-resolution and from lower paleolatitudes) are quite few. Considering so many uncertainties for Mg/Ca, we have thus used TEX86 data and no longer use Mg/Ca.

OK, now the age uncertainties. As far as I can tell, there isn't a consideration of this, but there needs to be. Based on the available chronological constraints for each event, one should be able to estimate the uncertainty simply by Monte Carlo-ing different possible age models that fall within the acceptable range. In "shallow-time" work I use BACON software (rbacon package in R) to do this with some Bayesian constraints, but even a simple MC version would be sufficient here. What you want to achieve here is an estimate of the uncertainty in time, which translates to the uncertainty in the rate of change of both temperature and extinction.

Response: Added. We have added the age uncertainty for each event by using Monte Carlo simulation as advised.

In the end, you will an ensemble of possible SST values from the proxies, and an ensemble of possible age models. Bootstrap sample across both to get a "final" estimate of uncertainty on your rates. Use this ensemble method to add error bars to all the figures, and for hypothesis testing.

Response: Added. We got the uncertainties of rates by bootstrapping SST values and ages and added these error bars to all figures (see Figs. 1-3).

Beyond the need for uncertainty quantification, my only other content suggestion concerns the K/Pg extinction, which is unique as it is associated with an impact. Hull et al 2020 (Science) showed that the volcanic outgassing and associated global warming near the K/Pg boundary predates the extinction and found no evidence that the extinction is related to the late Maastrichtian warming. The cooling that follows this warming, moving into the K/Pg, is relatively slow and small-magnitude, hence the authors conclude that it is the impact itself and associated climate changes that cause the extinction. Given the coarse resolution of the analysis in this paper, it is not clear to me whether the correlation analysis is reflecting the late Maastrichtian warming/cooling, or the volcanic winter that is

associated with the impact. From a cause-and-effect point of view, the analysis of the link b/t T and extinction for the K/Pg should be done specifically on the volcanic winter since that is the actual climate change associated with extinction.

Response: Yes, the K/Pg extinction event is indeed unique because it is associated with an impact. As suggested, the global warming below the K/Pg boundary has a small magnitude and predates the extinction, so is not what caused the extinction (Hull et al., 2020). The impact event is tightly linked with the extinction as they happened at the same time. In addition, TEX86 records show a significant and short-lived cooling in the “iridium anomaly” bed (Vellekoop et al., 2014, PNAS; 2016, Geology). This “impact winter” occurred following the latest Maastrichtian cooling. In our manuscript, we analyzed the link between extinction and the “combined” cooling (including the latest Maastrichtian cooling and the impact winter).

One more statistical note on the correlations: Reviewer 1 already brought this up, but the p-values on all correlations should account for the reduced degrees of freedom that comes from the autocorrelation in the time series plotted in Fig. 1. There are several methods to do this, my personal preference is to use the nonparametric method of Ebisuzaki et al. 1997 J. Climate ([https://doi.org/10.1175/1520-0442\(1997\)010<2147:AMTETS>2.0.CO;2](https://doi.org/10.1175/1520-0442(1997)010<2147:AMTETS>2.0.CO;2)). I see that the authors addressed this with first differences, but that doesn't always do the job, so I recommend checking with Ebisuzaki to be sure.

Response: We performed autocorrelation function (ACF) for the three studied time series (i.e., ΔT , rate and GF extinction rate). Autocorrelations have no significant difference from zero. Consequently, serial correlations have no or only limited effect on our correlation statistics.

Finally, regarding the T compilations, I agree with Reviewer 2 that it is too limited. In particular, the Cenozoic events are sparsely represented and there is no need for that because there are tons of data out there. For the Paleogene, the authors might benefit from using the deepMIP compilation for the PETM and the Eocene (Hollis et al 2019) which is more comprehensive than what is used here. There also seems to only be one site for the Neogene time intervals, but I suspect there is much more out there; there should be some UK37 data available for example (?). I think it is important to use as much data as possible, so that the analysis isn't biased by use of a single or only a few sites. I don't see that the authors have addressed this comment by Reviewer 2 head-on yet, but I think that they should.

Response: Thanks for this suggestion. We have now added as much temperature data as we can that meet the criteria for inclusion, i.e., with clear age model and from lower paleolatitudes. For example, we calculate the temperature change across the Permian-Triassic boundary based on conodont oxygen isotope data from four well-studied sections, and during the Early Toarcian interval based on brachiopod and belemnite

oxygen isotope data from five sections. As suggested, for Paleogene intervals, we have added more data by using the deepMIP compilation of Hollis et al. 2019 and other references, e.g., Wade et al., 2012, O'Brien et al., 2020. For Ng1 (early Miocene) interval, we didn't find further suitable data. Although there are more temperature data in this interval (including both TEX86 and UK37), most are from high latitude regions, e.g., ODP site 982 (Conte et al., 2006; Super et al., 2020), IODP site U1338 (Rousselle et al., 2013), U1356 (Hartman et al., 2018), ANDRILL AND-2A core, Ross Sea (Levy et al., 2016).

Here are a few line-by-line comments:

-Line 194: Perhaps update this discussion to SSP5-8.5 to reflect the AR6 scenarios.

Response: Revised. RCP 8.5 has been updated to SSP5-8.5.

-Line 208: belemnite Mg/Ca

Response: Revised.

-Line 214: BIT is probably not the best criterion, see above comments on treatment of TEX86.

Response: Thanks. Other criteria have been considered as discussed above.

-Line 226: Moving averages have the undesirable effect of "time-shifting" your data so I wouldn't use them. Use a Gaussian smooth instead if you would like to minimize sensitivity to outliers. Most statistics software (Matlab, R) have built-in functions for this. Apply the same method to all of your sites, rather than variously relying on moving averages, kernel regression, etc.

Response: Revised. We applied Monte Carlo simulation to get the magnitude and the rate of temperature change for all time intervals. Delta T and R and their uncertainties were calculated using the distribution of data from two groups around t0 and t1.

Good luck with the revisions and feel free to contact me if anything in the above is unclear.

-Dr. Jessica Tierney

Response: Thanks again for your constructive comments.

Reviewer comments, third round –

Reviewer #4 (Remarks to the Author):

Song et al have done a very thorough job of addressing my comments in the last revision. I really appreciate the attention to detail with respects to the proxy data and the uncertainty propagation, which I think makes the conclusions more robust. Just one tiny point of clarification: for d18O, no regional seawater corrections (i.e. by latitude, a la Zachos '94 for example) were made, as I understand it. If I've got that right, I would just state that in the d18O methods section for clarity.

Congratulations on the paper, I think this will be of great interest to many.

-Jessica Tierney, University of Arizona.

Note: original reviewer comments in black; responses of authors in blue

REVIEWER COMMENTS

REVIEWERS' COMMENTS

Reviewer #4 (Remarks to the Author):

Song et al have done a very thorough job of addressing my comments in the last revision. I really appreciate the attention to detail with respects to the proxy data and the uncertainty propagation, which I think makes the conclusions more robust. Just one tiny point of clarification: for d18O, no regional seawater corrections (i.e. by latitude, a la Zachos '94 for example) were made, as I understand it. If I've got that right, I would just state that in the d18O methods section for clarity.

Response: We appreciate the reviewer's very positive comments. For d18O, we have added the statement that "Most oxygen isotope data are from tropical and subtropical regions (Supplementary Fig. 5), therefore, no latitudinal seawater corrections were made.", see Lines 243-245.

Congratulations on the paper, I think this will be of great interest to many.
-Jessica Tierney, University of Arizona.

Response: Thanks again for your constructive comments.